# An atlas of the human liver diurnal transcriptome and its perturbation by hepatitis C virus infection

Atish Mukherji [1,23], Frank Jühling[1,23], Yogy Simanjuntak [1], Emilie Crouchet[1], Fabio Del Zompo[1], Yuji Teraoka[2], Alexandre Haller[3], Philippe Baltzinger [3], Soumith Paritala[4], Fahmida Rasha[4], Naoto Fujiwara[4], Cloé Gadenne[1], Nevena Slovic[1], Marine A. Oudot[1], Sarah C. Durand[1], Clara Ponsolles[1], Catherine Schuster [1], Xiaodong Zhuang[5,6], Jacinta Holmes[7], Ming-Lun Yeh[8], Hiromi Abe-Chayama[9], Mathias Heikenwälder [10,11], Angelo Sangiovanni[12], Massimo Iavarone[12], Massimo Colombo[13], Steven K. H. Foung[14], Jane A. McKeating [5,15], Irwin Davidson [3], Ming-Lung Yu [8,16], Raymond T. Chung[17], Yujin Hoshida [4], Kazuaki Chayama [18,19], Joachim Lupberger [1,24] ✉ & Thomas F. Baumert [1,20,21,22,24] ✉

Chronic liver disease and cancer are global health challenges. The role of the circadian clock as a regulator of liver physiology and disease is well established in rodents, however, the identity and epigenetic regulation of rhythmically expressed genes in human disease is less well studied. Here we unravel the rhythmic transcriptome and epigenome of human hepatocytes using male human liver chimeric mice. We identify a large number of rhythmically expressed protein coding genes in human hepatocytes of male chimeric mice, which includes key transcription factors, chromatin modifiers, and critical enzymes. We show that hepatitis C virus (HCV) infection, a major cause of liver disease and cancer, perturbs the transcriptome by altering the rhythmicity of the expression of more than 1000 genes, and affects the epigenome, leading to an activation of critical pathways mediating metabolic alterations, fibrosis, and cancer. HCV-perturbed rhythmic pathways remain dysregulated in patients with advanced liver disease. Collectively, these data support a role for virus-induced perturbation of the hepatic rhythmic transcriptome and pathways in cancer development and may provide opportunities for cancer prevention and biomarkers to predict HCC risk.

Mammals are endowed with an endogenous timing system known as the circadian clock (CC), a well-known regulator of physiology and behavior[1–4]. The CC is self-sustained and present in virtually all cell and tissue types. The CC-oscillator represents the core of this molecular "clock" and operates as a transcriptional-translational feedback system[1–6]. Notably, components of the CC-oscillator drive daily rhythmicity of their own expression and the temporal expression of target genes (CC-controlled genes; CCGs), thereby ensuring the coordinated function of cells, tissues, and organs[1–4]. Studies with animal model systems reveal an essential role for the CC to regulate metabolism, immune and endocrine functions[1–4,7]. Furthermore, perturbation of CC function in murine models associates with diverse

pathologies including metabolic diseases and cancer[1–4,7–11]. However, the role of disrupted liver CC in the development of human chronic liver disease is largely unknown due to our limited knowledge of diurnal gene expression in the human liver.

Hepatitis C virus (HCV) is a major cause of chronic liver disease and hepatocellular carcinoma (HCC). Direct-acting antivirals (DAA) can eliminate HCV and cure infection[12,13], however, a significant HCC risk persists, especially in patients with advanced fibrosis, even after viral elimination[12–17]. The lack of reliable biomarkers to predict HCC risk after viral cure poses a clinical challenge[15–17]. Despite significant strides, the molecular basis of HCV-induced liver disease and HCC development is still only partially understood[14,15]. Several studies suggest a role for HCV-induced transcriptional reprogramming in the development of HCC[18–21], however, the molecular drivers and mechanisms remain unknown. Here, we aimed to investigate the role of the hepatic clock in liver disease biology by identifying the diurnal transcriptome and epigenome of human hepatocytes and its perturbation in chronic HCV infection.

## Results

### Comparative analysis of the rhythmic transcriptome in human and mouse hepatocytes

Assessing the rhythmicity of the liver transcriptome in humans is challenging as it would require multiple biopsy samples over a 24 h period which poses a non-acceptable risk for the patient. We, therefore, used a human liver chimeric mouse (HLCM) model as a surrogate. These mice are immunodeficient and engrafted with primary human hepatocytes (PHH) and are a well-established model to recapitulate critical aspects of patient liver disease biology, including chronic viral and metabolic liver disease[20,22–24]. To identify the genes displaying a rhythmic expression pattern in human hepatocytes, we investigated temporal changes in transcript abundance in liver tissue samples from male HLCM (Fig. 1a). Hence, three HLCM were sacrificed every 4 h throughout the 24-h period starting at *Zeitgeber* 0 (ZT0; time of light ON, and ZT12- time of light OFF). We performed two independent animal experiments (Series 1 and 2; Supplementary Data 1, 2) to map the human hepatic diurnal transcriptome. Measurement of human serum albumin levels indicated comparable levels of hepatocyte engraftment (degree of humanization) in both experiments (Series 1: ~14,203 µg/mL, and Series 2: ~14,973 µg/mL; Supplementary Data 1). This was further confirmed by comparing human vs. mouse RNA-sequencing (RNA-seq) reads (Supplementary Data 2) and human hepatocyte-specific CK8-18 immunostaining (Supplementary Fig. 1), suggesting ~65% to ~70% humanization of the chimeric livers. Importantly, hematoxylin-eosin (HE) staining revealed a conserved lobular hepatic architecture in HLCM (Supplementary Fig. 1). To assess the temporal changes in the liver transcriptome of HLCM RNA-sequencing (RNA-seq) was performed, which produced an average of 35 million reads per sample (Supplementary Data 2). To distinguish transcripts originating from human and mouse tissue, sequence reads were mapped and annotated to both the human and mouse genome (Methods, and Supplementary Data 2). Post-annotation unsupervised clustering was performed to determine the biological heterogeneity amongst HLCM samples with respect to the expression phase of all the core CC-oscillator genes and CC-output regulatory transcription factors in human hepatocytes (Supplementary Fig. 2a, b). This unbiased approach allowed us to choose two samples from each timepoint from Series 1 and all three samples for each timepoint from Series 2 for subsequent evaluations (Supplementary Fig. 2a, b). One sample of Series 1 had to be excluded because of insufficient reads (Supplementary Fig. 2a; indicated in red). Taken together, we analyzed five HLCM per timepoint to ensure a robust dataset for further analysis of rhythmic transcriptome in human hepatocytes.

The rhythmicity of CC-gene expression in the chimeric livers were analyzed by *dryR*[25]. Notably, core CC-genes in engrafted human hepatocytes from both experiments showed comparable oscillation profiles to residual murine hepatocytes and showed the characteristic rhythmicity of these genes (Series 1 vs. Series 2 in Fig. 1b, and merged Series 1 and 2 in Supplementary Fig. 3a)[1–4]. These comprise the RORE-containing genes *CRY1*, *BMAL1*, *NPAS2*, *CLOCK*, and *NFIL3*, which reach maximal expression during the active phase (ZT12-ZT0) and minimal levels during the rest phase. In contrast, BMAL1/CLOCK-transcriptional activity-dependent E-Box containing genes *DBP*, *TEF*, *PER1*, and *PER2* are mostly expressed during rest phase (Fig. 1b and Supplementary Fig. 3a). Next, we compared the expression of CC-genes between HLCM liver and in the liver of wild-type (WT) mice[5,25] which suggested their phase advancement in the transplanted liver (Supplementary Fig. 3b). Importantly, our analyses of the expression pattern of the CC-genes in engrafted human hepatocytes, residual murine cells, and their comparison with the WT mice liver clock (Fig. 1b and Supplementary Fig. 3a, b) is consistent with recent observations[26]. Next, we applied the recently developed algorithm *dryR*[25] to analyze the differential rhythmicity of gene expression in the HLCM liver. This algorithm determines variations of amplitude (fold changes), phase (peak expression time), and mean expression levels of circadian orthologous gene expression, allowing the comparison of datasets with multiple conditions and between species. The *dryR* algorithm distributes groups of genes to different models of rhythmicity. Through *dryR* we identified five distinct categories of transcriptional profile in HLCM livers that include genes with cycling in human hepatocytes (model 2; colored green in the cartogram), cycling in mouse hepatocytes (model 3; pink in the cartogram), unaltered rhythm [model 4, gray: merger of pink (mouse) and green (human)], and altered rhythm (model 5; diverged expression between two species either in phase or amplitude while displaying some rhythmicity) (Fig. 1c and Supplementary Data 3). Additionally, a fifth model includes non-rhythmic genes in both species (model 1; Supplementary Data 3). *dryR* revealed ~1700 rhythmic protein-coding orthologous genes in human hepatocytes of HLCM liver (models 2, 4, and 5). This included 824 genes, which were uniquely rhythmic in human cells (model 2; Fig. 1c and Supplementary Data 3). We also found 103 genes (model 5) whose expression was altered between human and mouse hepatocytes, while 749 genes (model 4) showed unaltered rhythmic expression in two species in HLCM liver (Fig. 1c and Supplementary Data 3). Comparing cycling genes in human hepatocytes of HLCM with that of WT mice livers[5] and post-mortem human livers[27] revealed rhythmic genes which are shared (Supplementary Fig. 4a, b and Supplementary Data 3). However, the 'number' of shared genes were not statistically (hypergeometric test) significant, probably due to (i) whole liver tissue contains cells other than hepatocytes and (ii) using *dryR* we analyzed only protein-coding genes (with known murine orthologues) while different categories of non-coding genes are also known to display rhythmicity in whole liver tissue.

Next, a classification of rhythmic genes in human hepatocytes of HLCM for their biological function using the Molecular Signature Database (MSigDB)[28,29] allowed a comparison with known 'clock'-controlled physiological processes reported in the mouse liver[5,25] (Fig. 1d and Supplementary Fig. 5a, b). Temporally resolved enrichment analysis revealed an overwhelming role for the hepatic clock to regulate pathways involved in leptin, bile acid, cholesterol, fatty acid, and heme metabolism. In addition, we found that human rhythmic pathways involved in stress response (DNA repair, reactive oxygen species, p53 signaling), signaling pathways (NOTCH, KRAS, TGF-β), and cell cycle clustered in comparable temporal windows ("peak" expression ± 1 ZT) as seen in mice (Fig. 1d and Supplementary Fig. 5a, b). Human hepatocytes showed a rhythmic expression of cellular processes (apoptosis, hypoxia response), cytokine signaling (IFNα, IFNγ, IL6, TNFα, complement system), VEGF (angiogenesis), and unfolded protein response which operates at different temporal windows compared to

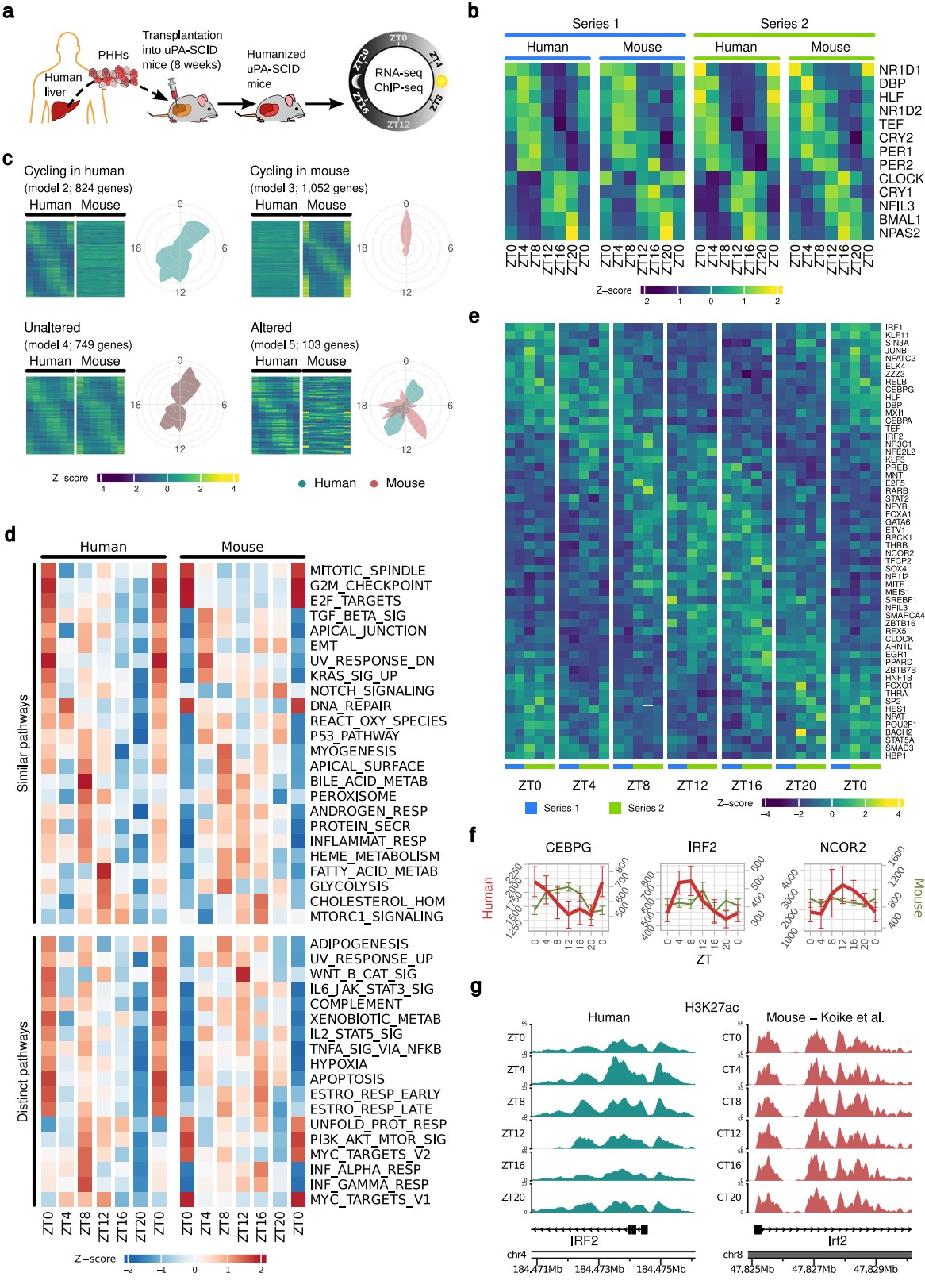

murine hepatocytes. Consistent with a recent study[26], we also observed that some of the key genes involved in growth hormone (GH)-STAT5[30] and mTOR signaling pathways, e.g., *STAT1, STAT3, STAT5B, MTOR*, and *LAMTOR1-5* were non-rhythmic (model 1) in transplanted human hepatocytes (Supplementary Data 3). Taken together, our study identifies the diurnal transcriptome and rhythmic physiological processes in human hepatocytes.

## Diurnal regulation of transcription factors and enhancer regions impact pathways relevant to liver homeostasis

Rhythmic expression of transcription factors (TFs) can generate temporal patterns of target gene expression. To identify rhythmically expressed TFs in human hepatocytes of HLCM, we analyzed a dataset containing 1600 human TFs[31]. Analyses of *dryR*-classified rhythmic genes led to the identification of ~140 rhythmically expressed TFs

**Fig. 1 | The rhythmic transcriptome of human hepatocytes in vivo. a** Schematic representation for the generation of immunodeficient humanized liver chimeric mice (HLCM). Liver tissues obtained from these HLCM were used to perform RNA-seq and ChIP-seq. PHH primary human hepatocyte. The cartoon was created by the authors using images from freely available online resources. **b** Mean rhythmic expression pattern of core CC-oscillator genes and CC-output regulators in human and mouse liver cells of HLCM in two independent experiments (Series 1 and 2). Series 1: $n = 2$/timepoint, Series 2: $n = 3$/timepoint. Source data are provided as a Source Data file. **c** Mean expression pattern of genes showing rhythmicity in human and mouse hepatocytes in HLCM as predicted by *dryR* ($n = 5$ HLCM/ diurnal timepoint). *dryR* identified four models of rhythmic genes in HLCM. A fifth model comprising only non-cycling genes is not shown. Phase distribution of respective models is indicated by radial coordinates, where green-human, pink-mice, and gray-overlap of humans and mice. Source data are provided as a Source Data file.

**d** HALLMARK pathways significantly (FDR <0.05) enriched for rhythmic genes in human hepatocytes in HLCM ($n = 5$ HLCM/ timepoint) as listed in (**c**) and their expression in WT mice. Similar pathways show overlapping (±1 ZT step) of peak enrichment scores comparing human and WT mice maximum enrichment scores. Source data are provided as a Source Data file. **e** Rhythmic **e**xpression of transcription factors (TFs) in human hepatocytes in HLCM ($n = 5$ HLCM/ timepoint), as predicted by *dryR*. Source data are provided as a Source Data file. **f** Examples of the mean expression pattern of *dryR* identified TFs as listed in (**e**). The Y-axis represents the DESeq2 normalized reads. Human hepatocytes (red) and residual murine (green) cells. $n = 5$ HLCM/timepoint. Bars represent SD. Source data are provided as a Source Data file. **g** ChIP-seq coverage plots indicate diurnal variations in H3K27ac levels in *IRF2* in human hepatocytes and WT mice liver. Human (green) and murine (red) cells. ZT0: represented twice in (**b**–**f**) in each panel to maintain conformity.

(~8% of all rhythmic genes) in human hepatocytes of HLCM livers (Fig. 1e and Supplementary Figs. 6, 7a, b) that are well-known regulators of physiopathology. Evaluating the expression of TFs in human and mouse cells of HLCM revealed that key TFs like *IRF2, NCOR2, JUNB, RELB*, and *IRF1* were uniquely rhythmic in human hepatocytes (model 2; Fig. 1f, Supplementary Fig. 6, and Supplementary Data 3), correlating with the pathway analysis indicating that several of the cytokine and stress response pathways are temporally distinct in human and mouse livers (Fig. 1d). We found shifted expression phase of TFs regulating xenobiotic metabolism (*NR1I2*, and *HLF;* Supplementary Figs. 6, 7a) and altered expression (model 5) for *CEBPG* (Fig. 1e, f). Importantly, amongst rhythmically expressed TFs in both species in HLCM liver contained CC-regulatory TFs (*BMAL1/ARNTL*, *CLOCK*, *NR1D1*), and orchestrators of metabolism (*PPARD* and *CREB3L3*). In HLCM livers, we identified several human orthologues of zinc finger TFs (ZNFs) being expressed in different *dryR* categories, e.g., *ZNFs* (*217*, *248*, and *318*; model 2 - *only cycling in human hepatocytes*), while *ZNFs* (*330*, *277*, *362* and *367*, model 4 - *unaltered in two species*) (Supplementary Fig. 6 and Supplementary Data 3).

Gene expression requires epigenetic changes at promoter and enhancer regions. To determine whether these rhythmicity in transcript levels are associated with diurnal epigenetic remodeling of the chromatin, we performed ChIP-sequencing (ChIP-seq) of samples from the HLCM liver (Fig. 1a). We initially focused on the temporal variation in histone 3 lysine 27 acetylation (H3K27ac) levels an established marker of promoter and enhancer activation[32,33], and whose dysregulation is related to chronic liver disease[19,34,35]. Comparative analysis revealed the rhythmicity of H3K27ac levels in enhancer and promoter regions in human hepatocytes and in the liver of WT mice[5]. Analyzing the enhancer activation pattern showed that the changes for CC-output regulatory TF D-box binding protein (*DBP*) follow a similar pattern in human and murine hepatocytes, while variations in H3K27ac levels surrounding *IRF2* promoter-enhancer were only rhythmic in the human hepatocytes (Fig. 1g and Supplementary Fig. 8).

Taken together, our results provide an atlas of the human hepatocellular rhythmic transcriptome, unveiling its epigenomic variations in vivo. Importantly, altered expression of several of the identified rhythmic genes and processes are reported to be associated with chronic liver disease[19,20,35].

## HCV infection perturbs the rhythmicity of human liver transcriptome in vivo

To understand whether perturbation of the human hepatocellular rhythmic transcriptome associates with liver disease and cancer, we investigated the impact of HCV infection on diurnal gene expression in two independent animal experiments (Series 1 and 2; Supplementary Data 1, 2). HCV exclusively infects human hepatocytes[12,13,17] and does neither infect nor replicates in murine cells[22,24,36]. HLCM were infected with patient-derived HCV for 10 weeks (Fig. 2a and Supplementary

Data 1) and assessed for hepatic diurnal gene expression. RNA-seq from the HCV-infected HLCM livers (3 HLCM/experiment/diurnal timepoint) yielded specific transcriptomic profiles that clustered in an unsupervised manner (Supplementary Data 2 and Supplementary Fig. 9a, b). While humanization of the mouse livers in control and HCV animals were similar at the time of infection according to human albumin levels in the sera, we observed a slightly reduced humanization in the HCV animals after 4 weeks of infection (Supplementary Fig. 10c) as described previously in a similar model[24] (Supplementary Fig. 10a–c and Supplementary Data 1). Loss of human hepatocytes was likely due to an induction of cellular stress in HCV-infected livers[19,20,24,37], which was also reflected histologically (Supplementary Fig. 10b). We also measured viral load from the serum of HCV-infected animals (Supplementary Fig. 11). Importantly, we employed a species-specific 'reads' mapping strategy which allows for an independent analysis of human and mouse transcriptome (see methods). Additionally, this method of analyzing humans (and mouse) reads independent from each other also considers artifacts arising from different humanization levels in different groups of HLCM. Due to the restricted tropism of HCV to human hepatocytes, we studied the impact of HCV infection on diurnal transcriptome only in human hepatocytes (control and infected) of HLCM.

In the human hepatocytes of HLCM, we investigated HCV-induced perturbation of the rhythmic transcriptome and pathways using *dryR*[25]. As shown above (Fig. 1c), we identified five distinct categories of transcriptional profiles in HCV-infected HLCM livers (loss, gain, unaltered, and altered rhythm (Fig. 2b and Supplementary Data 4), plus one additional category of non-rhythmic genes in comparison to non-infected controls. The global impact of HCV-induced perturbation of the hepatocellular diurnal transcriptome (~22% of all the rhythmic genes; models 2, 3, and 5) was reflected by the dysregulation of several physiological processes (Fig. 2c and Supplementary Fig. 12), including pathways of HCV-induced proteogenomic changes[20,38,39]. Importantly, pathways (enriched for cycling genes; FDR < 0.05) being significantly dysregulated comprise key drivers of chronic liver disease, including metabolic alterations (fatty acids, lipids, peroxisome organization), fibrosis (TGFβ-signaling, SMAD activity, fibroblast proliferation, and EMT response), and oncogenic pathways linked to HCC development (liver cancer signatures, MYC, H-RAS, and EGFR signaling) (Fig. 2c, Supplementary Fig. 12). All these pathways were previously determined by *dryR* to be cycling (Fig. 1d) within model 2 (loss of rhythmicity) (Fig. 2c). Although HLCM lack T- and B-cells (adaptive immunity deficient), engrafted hepatocytes can activate cell-intrinsic innate immunity-related pathways[19,20], which may act to defend against stress and infection. Consistently, HCV-infected HLCM liver showed an upregulation of innate immune and inflammatory pathways (NF-κB, TNFα, IL6/STAT3, and Type I interferons) (Fig. 2c and Supplementary Fig. 12). Next, we performed immunostaining of MYC, a well-known oncogene driving various aspects of chronic liver disease and HCC.

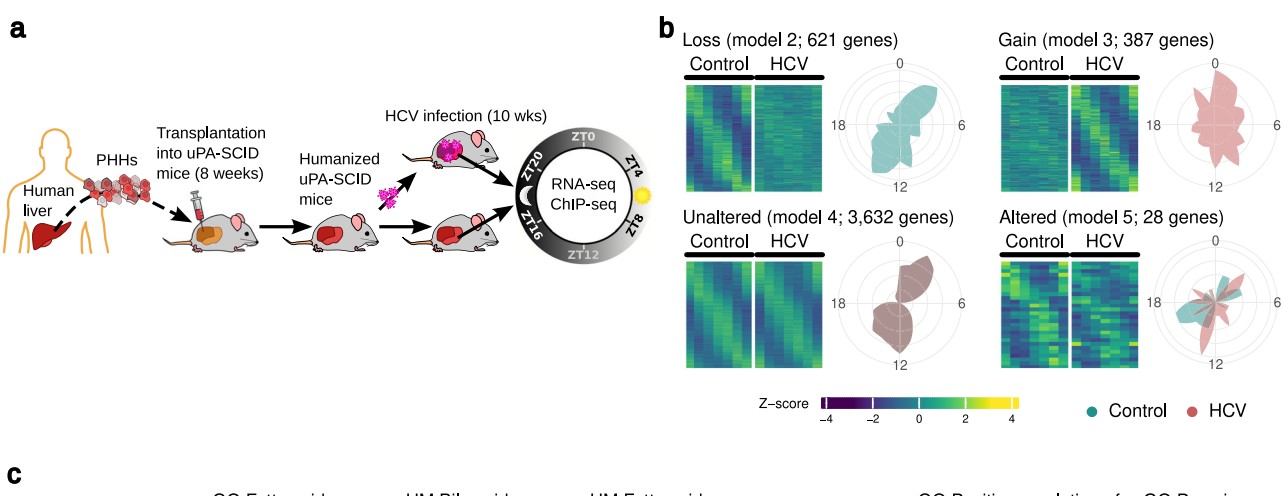

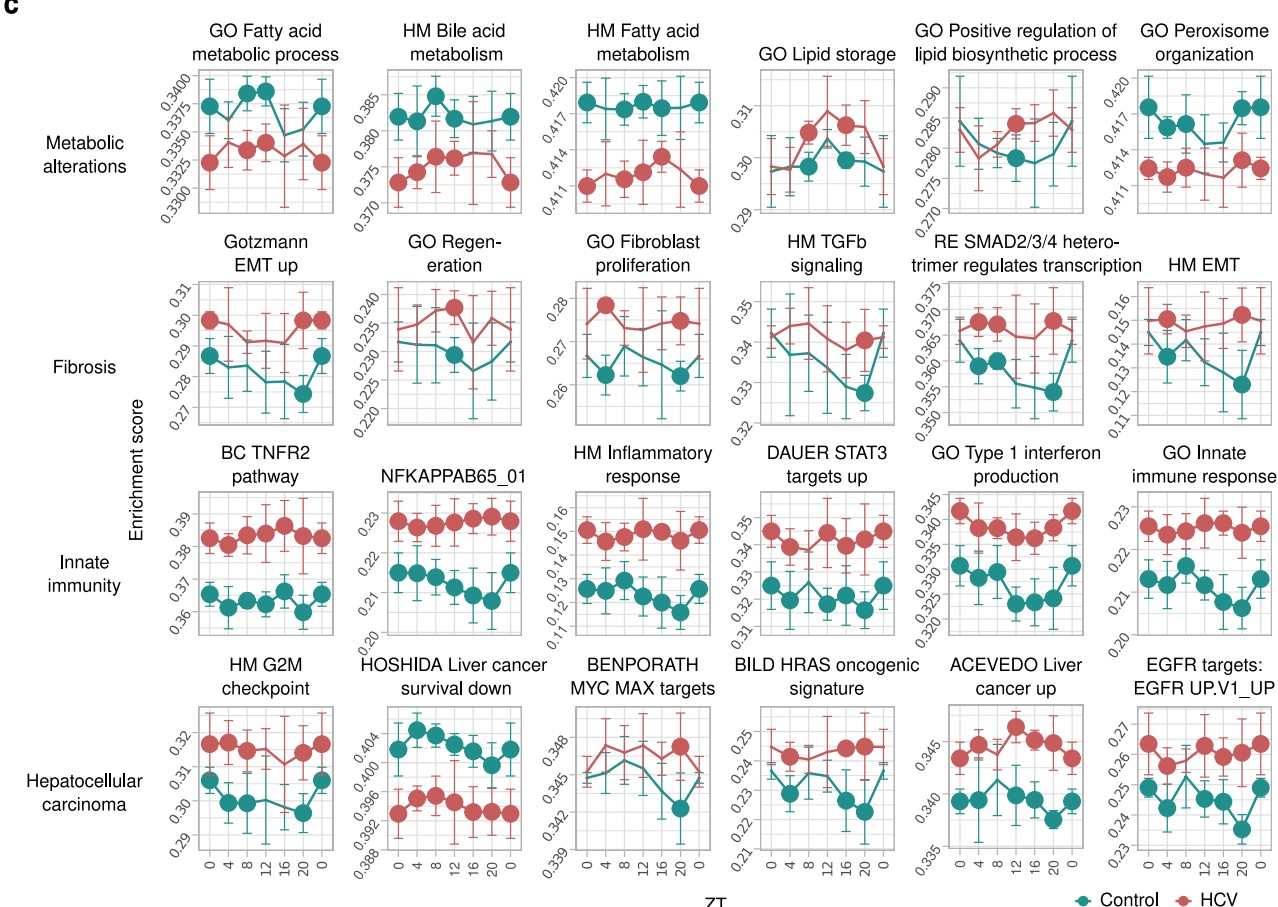

**Fig. 2 | Chronic HCV infection disrupts the rhythmicity of human liver transcriptome in vivo. a** Schematic representation of the chronic HCV infection of HLCM. Liver tissues obtained from these infected HLCM were used to perform RNA-seq and ChIP-seq. Two independent infection experiments were performed. PHH primary human hepatocytes. The cartoon was created by the authors using images from freely available online resources. **b** Four categories of CC-disturbed and unaltered genes identified by *dryR* according to their expression in control and HCV-infected human liver cells in HLCM ($n = 5$ HLCM/group and timepoint). A fifth model comprising only non-cycling genes is not shown. Phase distribution of indicated models is indicated by radial coordinates, where green: control, pink:

HCV, and gray: overlap of control and HCV. Source data are provided as a Source Data file. **c** Examples for mean enrichments over indicated timepoints of pathways significantly enriched (FDR < 0.05) for rhythmic genes (i.e., HM HALLMARK, GO Gene Ontology, BC BIOCARTA, RE Reactome gene sets; see Fig. 1d), shown for controls and HCV-infected livers. $n = 5$ HLCM/group and timepoint. Bold dots inside figure panels represent significant differences in enrichments ($p < 0.05$; Wilcoxon signed-rank test, two-tailed). Bars represent SD. Source data, including the exact $p$ values are provided as a Source Data file. ZT0: represented twice in (**b**, **c**) in each panel to maintain conformity.

While MYC was barely detectable in the uninfected liver, it was significantly upregulated upon HCV infection in the HLCM liver, supporting the activation of downstream target pathways at the transcriptomic level (Fig. 2c and Supplementary Fig. 13a–c). Several studies have noted that the lack of adaptive immune cells prevents the

development of steatosis and that only marginal fibrosis develops in HLCM livers[37,40,41]. Consistently, we did not detect steatosis in the histology of HLCM livers (Supplementary Fig. 14a). Sirius red staining and quantification of collagen-positive areas (CPA) showed modest but significant liver fibrosis in the infected mice (Supplementary

Fig. 14b, c). In summary, these data show that HCV infection dysregulates the diurnal liver transcriptome and rhythmic processes, which likely predisposes the advancement of chronic liver disease.

## HCV infection disrupts rhythmic epigenetic variations that drive gene expression in vivo

The diurnal variation of epigenomic changes driving promoter-enhancer activation/repression in health and chronic liver disease in human hepatocytes is unexplored. Hence, we profiled 24-h changes in two epigenetic markers of gene activation: histone 3 lysine 27 acetylation (H3K27ac) and histone 3 lysine 9 acetylation (H3K9ac) in the liver obtained from infected and non-infected HLCM. We identified global rhythmicity in both H3K27ac and H3K9ac levels in promoter-enhancer regions of cycling genes in uninfected livers (Fig. 3a, d and Supplementary Data 3). Amongst cycling genes, we identified transcription start site (TSS)-enriched peaks for both H3K27ac and H3K9ac. Moreover, we also found peaks of H3K27ac and H3K9ac within gene bodies and intergenic regions that suggests possible diurnal epigenetic regulation of yet-to-be-characterized distal enhancer regions (Fig. 3a–f and Supplementary Data 5). Importantly, the overall diurnal variation in H3K27ac was completely lost in the infected liver. The loss of H3K27ac variation emerged as a generalized plateau of TSS-associated H3K27ac peaks (Fig. 3a–c). Also, the observed diurnal intergenic enhancer regions were persistently hyper-H3K27 acetylated in the infected liver (Fig. 3a). In contrast to H3K27ac, HCV-induced higher H3K9ac levels mostly from the end of the rest phase and during the active phase that caused a saturation of peak numbers (ZT8-ZT20) when compared to the non-infected animals (Fig. 3d–f). Thus, HCV alters diurnal epigenome at both promoter-enhancer (H3K27ac and H3K9ac), and gene body (H3K9ac) levels (Fig. 3a–f). In addition, analyses of gene-specific diurnal variations in levels of H3K27ac and H3K9ac of liver disease-driving genes (*XBP1* and *RAF1*) and CC-component *CRY1* confirmed the HCV-induced epigenetic perturbations (Fig. 3g and Supplementary Fig. 15a, b). To unravel the TF motifs most enriched in H3K27ac peaks in HCV-infected HLCM livers, we intersected TSS and enhancer-associated peaks with TF binding site predictions from JASPAR[42,43] (Fig. 3h). This revealed an overall suppression of H3K27ac levels in the regulatory regions of genes controlled by CC-output regulatory TFs (e.g., DBP, HLF, TEF, NFIL3; regulators of D-Box genes). Importantly, we also observed an enrichment of H3K27ac peaks in enhancer regions for known transcriptional drivers of chronic liver disease progression in the HCV-infected HLCM, e.g., NRF1/NFE2L1 (regulator of lipid metabolism, oxidative/hypoxic stress), E2Fs and EGRs (regulator of the cell cycle, stress response) (Figs. 2c, 3h). Taken together, our results demonstrate that HCV infection perturbs the hepatic transcriptome and epigenome to create an environment that is conducive to the development of liver disease and cancer.

## HCV perturbation of the diurnal gene expression is associated with HCC risk in patients with chronic liver disease

To investigate the clinical impact of these findings, we investigated the virus-induced perturbation of the rhythmic pathways in patients with chronic hepatitis C. Our analysis of transcriptomic changes in these patients supports our observations in HLCM by revealing that chronic HCV infection perturbs the expression of liver disease-relevant rhythmic pathways (Fig. 4a), as shown in a side-by-side comparison in this well-characterized cohort of pooled transcriptomic data from chronically HCV-infected patients and control subjects without liver disease[17] (Fig. 4a). Most of the liver specimens in these patient cohorts were obtained in the morning, however, we cannot exclude a moderate temporal heterogeneity due to a variation in clinical sampling times.

Next, we studied whether DAA cure results in the reversal of transcriptomic changes of the evaluated pathways in the HLCM mouse model (Control vs HCV-infected and HCV-infected vs DAA-treated; Supplementary Fig. 17a) as well as in HCV-infected and DAA-cured patients (Supplementary Table 1 and Supplementary Fig. 17b, c). Independent analyses of these cohorts revealed that the virus-induced changes in the human transcriptome are only partially reversed following viral clearance in patients with advanced liver disease and cancer (Supplementary Fig. 17b, c). Interestingly, we observed a perturbation of these pathways also in an independent patient cohort with MASH (Supplementary Fig. 17d).

Finally, we aimed to study whether the perturbation of rhythmic gene expression was associated with HCC risk in patients. We have previously shown that HCV-induced epigenetic changes are associated with a persistent HCC risk post viral cure[35]. The effect on carcinogenic pathways is reflected by the perturbation of the well-characterized prognostic liver signature (PLS)[21,38,39], which robustly predicts survival, liver disease progression, and HCC risk in patients with different etiologies, including chronic hepatitis C. Previous studies have shown that the PLS is perturbed to an HCC high-risk status during chronic HCV infection[18,19]. A systematic dissection of the 186 PLS genes revealed that genes associated with a low HCC risk and good prognosis are cycling into a much higher proportion compared to the expression of HCC high-risk/poor prognosis genes (Fig. 4b). Thus, HCV-induced transcriptomic-perturbation seems to affect the expression good prognosis genes much more than the poor prognosis genes (Supplementary Fig. 18).

Next, we aimed to identify a gene signature that reflects HCV-induced perturbation of rhythmic genes and, at the same time, predict patient outcomes. Therefore, we assessed whether diseased human livers express the various *dryR* gene sets we identified in the HCV-infected HLCM liver (Fig. 2c). We compared the perturbed rhythm associated with disease phenotype by using our previously described genome-wide transcriptome profiles of liver tissues from 216 early-stage (Child-Pugh class A) cirrhosis patients[18]. We defined gene sets of HCV-infected and control hepatocytes for "early" (ZT0-ZT8) and "late" (ZT12-ZT20) diurnal timepoints, as upregulated genes with fold changes >4 in each condition. We observed a correlation of several gene sets from early timepoints of HCV-infected hepatocytes with several disease severity-related features as well as the presence of the HCC high-risk status of the PLS[18,21,38,44–47] (Fig. 4b, Supplementary Table 2, and Supplementary Data 6). In contrast, the gene sets derived from the non-infected hepatocytes and late timepoints in HCV-infected hepatocytes were associated with a low-risk status of the PLS. Using these gene sets as a common signature termed HCV$_{CLOCK}$, we were able to classify the HCC patients into high-risk (poor outcome) and low-risk (good outcome) groups. Importantly, we observed changes in the gene set enrichment index (GSEI), which indicates a "magnitude of correlation" with overall survival according to HCV$_{CLOCK}$ classification (Fig. 4c, d), suggesting possible clinical relevance. Finally, our analyses of HCV-disrupted rhythmic pathways in HLCM and patient cohorts revealed a firm link with all major 'hallmarks' of cancer (Fig. 4e). Collectively, our analyses demonstrate that the HCV-induced perturbation of the human hepatocellular rhythmic transcriptome correlates with clinical risk of developing HCC.

## Discussion

The liver is the central metabolic organ in mammals and performs evolutionary conserved metabolic processes. However, given an evolutionary distance of ~80 million years with nocturnal rodents in which the CC has been extensively studied, it is important to identify genes and corresponding pathways that are rhythmic in human tissues and hepatocytes, and which are deregulated in liver disease. Our study provides a comprehensive temporal gene expression atlas and identifies epigenomic changes in human hepatocytes that facilitate bioinformatic analyses to study their potential role in liver disease. The abundance and complexity of transcripts in a multifunctional tissue

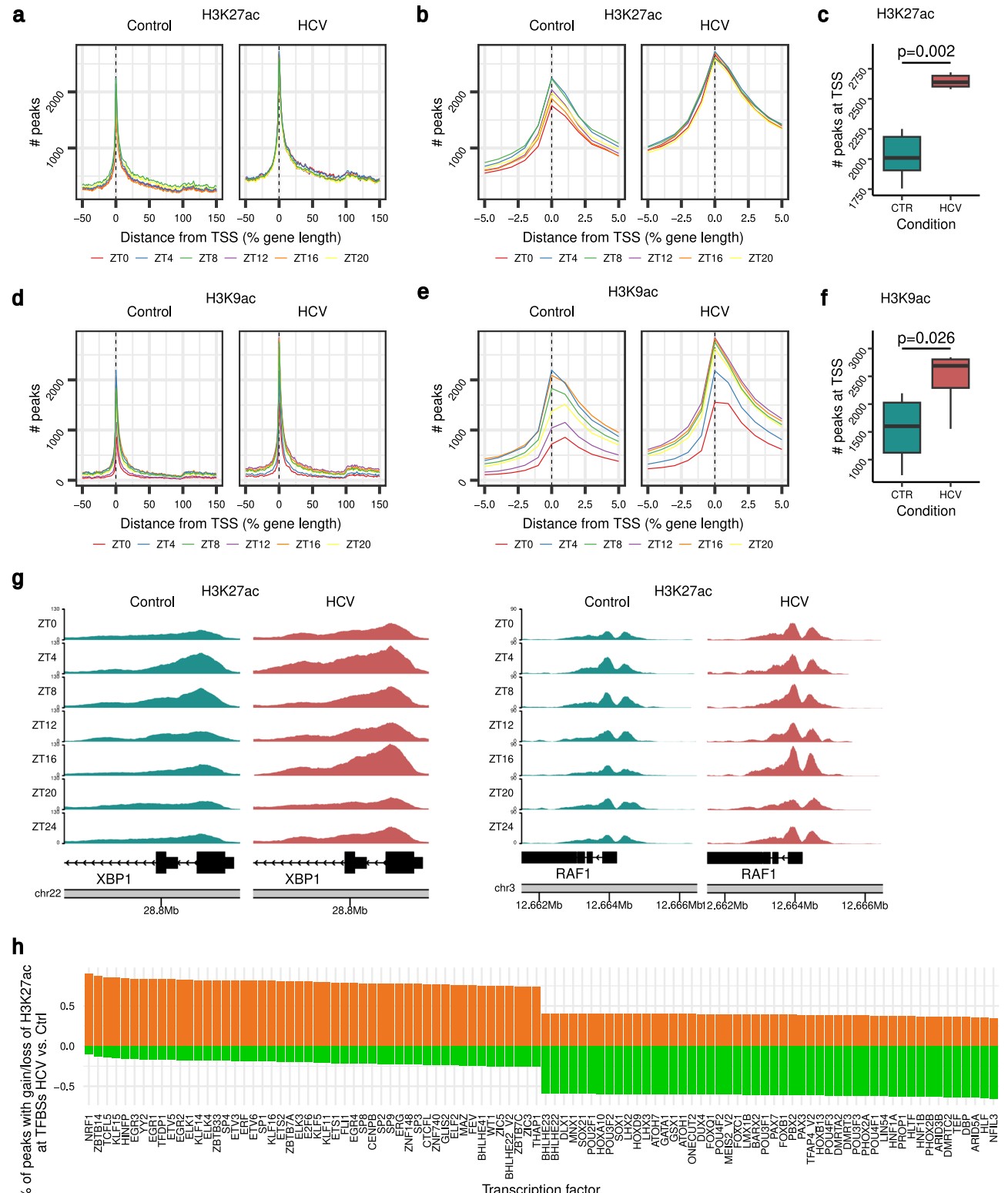

like the liver does not explain all dimensions of its physiology, but the study of temporal gene expression patterns, as provided here, offers a new and crucial dimension of understanding the human liver in health and disease. It has generally been assumed, given the conservation of biological processes across mammals, that diurnal transcriptomes are comparable across species while expressing mostly orthologous rhythmic genes. However, this investigation identifies crucial genes (TFs, epigenetic remodelers, enzymes, etc.) and pathways which show rhythmic expression only in human hepatocytes (Fig. 1 and

Supplementary Figs. 5–7). Critically, many of these pathways were recently identified as human hepatocyte-specific[26]. We found that the CC-genes in engrafted human hepatocytes and resident murine hepatocytes display comparable patterns of expression (Fig. 1b). This effect was recently confirmed and suggested to be due to a dominant cellular 'clock' synchronizing ability of engrafted human hepatocytes[26]. Together, our investigation and that of Delbés and co-workers[26] confirm that the engraftment of human hepatocytes into a new environment (neuronal and systemic signals, food, light cycle,

**Fig. 3 | Chronic HCV infection impairs the remodeling of the temporally regulated enhancers in vivo. a** Density of H3K27ac peaks at different timepoints across the gene bodies in control and HCV-infected human liver cells in HLCM. TSS transcription start site. Source data are provided as a Source Data file. **b** Density of TSS-associated H3K27ac peaks at different timepoints in control and HCV-infected human liver cells in HLCM. TSS transcription start site. Source data are provided as a Source Data file. **c** Quantitation of H3K27ac peak numbers at TSSs from (**b**) CTR control (green), HCV HCV-infected (red), $p = 0.002$, Mann–Whitney test, two-tailed, $n = 5$/timepoint. The box plots show the median (line), the 25th and 75th percentiles (box), and values within 1.5 times the interquartile range (whiskers). Source data are provided as a Source Data file. **d** Density of H3K9ac peaks at different timepoints across the gene bodies in control and HCV-infected human liver cells in HLCM. TSS transcription start site. Source data are provided as a Source Data file. **e** Density of TSS-associated H3K9ac peaks at different timepoints in control and HCV-infected human liver cells in HLCM. TSS transcription start site. Source data are provided as a Source Data file. **f** Quantitation of H3K9ac peak numbers at TSSs from (**e**), CTR control (green), HCV HCV-infected (red) $p = 0.026$, Mann–Whitney test, two-tailed, $n = 5$/timepoint. The box plots show the median (line), the 25th and 75th percentiles (box), and values within 1.5 times the interquartile range (whiskers). Source data are provided as a Source Data file. **g** Promoter-enriched H3K27-acetylation for HCV host factors *XBP1* and *RAF1* in control (green) and HCV-infected (red) human liver cells in HLCM. **h** Percentages of enhancer/TSS-enriched peaks with gain (orange) or loss (green) of H3K27-acetylation that overlap with transcription factor binding sites in gene targets as listed in the Jaspar database. Only the 50 top hits are shown for each direction. Source data are provided as a Source Data file.

temperature) did not prevent the re-establishment of a functioning rhythmic hepatic CC-oscillator. The HLCM model is well established to investigate viral and metabolic chronic liver disease. However, the CC-oscillator in engrafted human hepatocytes was phase-advanced when compared to the WT mice liver[26]. This is consistent with the literature in the field suggesting that this being due to the differential sensitivity of human hepatocytes to metabolic and systemic signaling[26,48,49]. It should be noted that HLCM are immunodeficient, and transcripts of liver resident immune cells cannot be studied extensively using such animal models. However, the engrafted hepatocytes have an intrinsic capacity to induce innate immune responses to stress and infection, as evidenced by their expression of several cytokines (TNF, IL6, and type I interferons) signaling pathways (Fig. 2c).

Using chronic hepatitis C as a model disease, our data indicate a critical role of the dysregulated hepatocellular rhythmic transcriptome for virus-induced cancer as shown by perturbation of the oscillation of the major cancer hallmark pathways in both HLCM and patients (Figs. 2–4). Notably, HCV infection was associated with a dysregulation of the transcriptomic oscillation of ~22% of rhythmic genes (loss, gain, and altered rhythmicity), and key biochemical pathways (Fig. 2 and Supplementary Fig. 12). Previous studies have demonstrated that viral infections and chronic liver disease is driven through similar perturbed pro-oncogenic pathways in various disease etiologies[19,21,38,50,51]. Our mechanistic studies identified a previously unrecognized epigenetic imprinting (dysregulated histone acetylation) in enhancer regions of these rhythmic genes, that correlated with impairments of cancer hallmark pathways upon viral infection (Fig. 3). Specifically, an HCV-induced perturbation of the promoter-enhancer-associated histone mark (H3K27ac) was present in vivo, while gene-body associated H3K9ac was only disrupted during ZT8-ZT20 (Fig. 3a–h). Our data indicate a dynamic interaction between different chromatin remodelers regulating gene expression at the epigenomic level. Interestingly, HCV infection did not perturb the transcript levels of major histone acetylases and deacetylases (Supplementary Fig. 16). However, additional in vivo validation and loss-of-function experiments would be required to unravel the detailed mechanistic events and cellular drivers in the temporal regulation of liver disease-relevant pathways.

A strength of our study is the use of the state-of-the-art in vivo model system for HCV infection based on male human liver chimeric mice with data integration of liver tissues of HCV-infected patients. A limitation of the study is that these model systems only partially mimic virus-host interactions in patients, and a diurnal analysis of the patient's liver transcriptome and proteome is not available.

Finally, our findings of gene expression in patients suggest a profound effect of HCV infection on HCC risk biomarkers (i.e., PLS) (Fig. 4b–d). As CC-targeting compounds[2] have been shown to attenuate liver disease progression in rodent models[52], our results may provide opportunities to discover CC-based biomarkers to predict HCC risk and novel therapeutic targets for cancer prevention.

## Methods
### Ethical statement
This study fully complies with ethical regulations regarding animal experimentation and analyses of patient samples. Animal protocols described in this study were performed in accordance with the Guide for the Care and Use of Laboratory. Animals (https://grants.nih.gov/grants/olaw/guide-for-the-care-and-use-of-laboratory-animals.pdf) and the experimental protocol was approved by the Ethics Review Committee for Animal Experimentation of the Graduate School of Biomedical Sciences, Hiroshima University with approval number A14-195. The study protocol for human patient samples was approved by the Hiroshima University Hospital ethical committee (approval number HI-98-21) in accordance with the Helsinki Declaration. All patients provided written informed consent.

### Generation of uPA/SCID humanized liver chimeric mice (HLCM) and HCV infection
Humanized liver chimeric mice were produced by splenic injection of cryopreserved PHH (BD Biosciences, San Jose, CA) into uPA/SCID mice[53]. The PHH donor was a deceased 2-year-old female patient with no recorded history of liver disease and viral infection. The isolated PHH showed positive enzymatic activities for multiple human cytochromes (CYP). PHH were stored and thawed as instructed to maintain their high viability (~75%). Both control and HCV-infected mice were engrafted with the same batch of PHH into the spleen. The uPA/SCID mice were obtained from PhenixBio: PXB-mouse [Genotype/Strain: cDNA-uPAwild/+/SCID, cDNA-uPAwild/+: B6;129SvEv-Plau, SCID: C.B-17/Icr-scid/scid Jcl]. All animal protocols described in this study were performed in accordance with the Guide for the Care and Use of Laboratory. Animals (https://grants.nih.gov/grants/olaw/guide-for-the-care-and-use-of-laboratory-animals.pdf) and the experimental protocol was approved by the Ethics Review Committee for Animal Experimentation of the Graduate School of Biomedical Sciences, Hiroshima University with approval number A14-195. All mice (male) were kept in isolated cages and fed ad libitum on a CFR1 diet (Oriental Yeast Co., Ltd, Tokyo, Japan). Twelve weeks after hepatocyte transplantation, mice (21 weeks old) were injected intravenously with or without $10^5$ copies of genotype 1b HCV-infected serum[54]. After serum inoculation, mouse blood samples were obtained serially, and human albumin concentrations and serum HCV RNA levels were measured. The serum sample was divided into small aliquots and stored at $-80\,°C$ until use. Both control and HCV-infected mice were provided ad libitum chow diet and water. Both groups of mice were maintained in 12 h light-dark cycle at 23-25 °C with humidity of 40-60% for 10 weeks and were sacrificed through carbon dioxide every four hours. Samples from HCV-infected and DAA-cured humanized liver chimeric mice published previously were all sacrificed during the circadian rest phase[35] (ethical approval from Hiroshima University Hospital; E-1049-1).

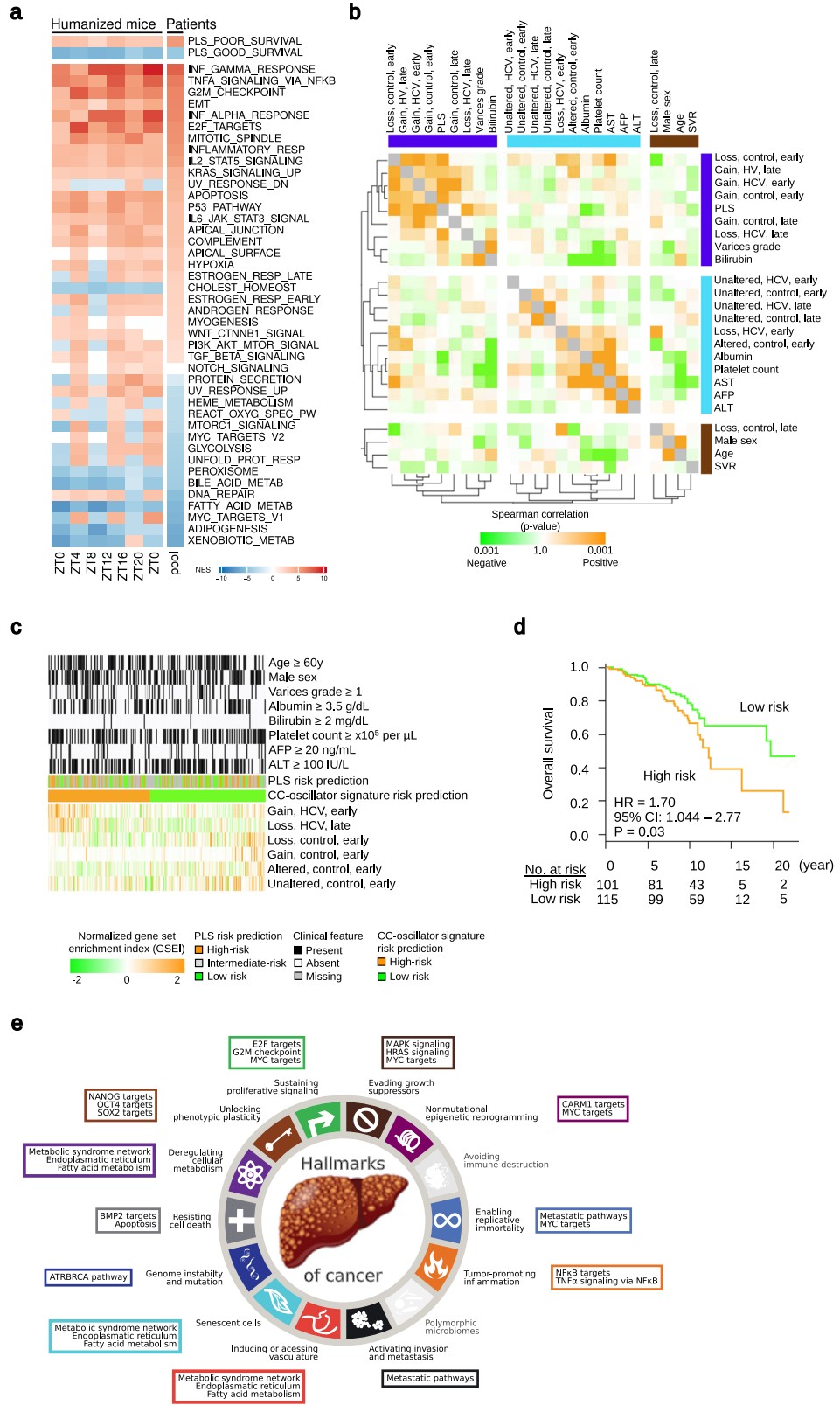

## Measurement of human albumin levels in the serum of human liver chimeric mice

Human hepatocyte repopulation rates were estimated by human serum albumin levels. Blood samples (5 μL) were collected periodically from the tail vein, and the levels of human albumin levels were determined by immunonephelometry using a JEOL BM6050 auto-analyser (JEOL, Tokyo, Japan) and LX Reagent Eiken Alb II (Eiken Chemical, Tokyo, Japan). Human serum albumin concentrations are expressed as ng/mL.

## Measurement of HCV load from serum samples

RNA was extracted from mouse serum samples using Sepa Gene RV-R (EIDIA Co., LTD., Tokyo, Japan). Extracted RNA was reverse transcribed using a random primer (TakaRa Bio Inc., Shiga, Japan) and M-MLV

**Fig. 4 | Chronic HCV infection perturbs the expression of rhythmic pathways in patients and cancer risk. a** Mean enrichments of Hallmark pathways enriched for human rhythmic genes as identified in Fig. 1d and the PLS[38] in the liver of HCV-infected HLCM and HCV-infected patients. Humanized mice: HCV-infected HLCM (Fig. 2; HCV vs. controls), Patients: HCV-infected vs. controls[17]. Pool: bulk analysis of all samples. Red=significant (FDR <0.05) enrichment, blue=significant (FDR <0.05) negative enrichment, white=no significant enrichment/unchanged. Source data are provided as a Source Data file. **b** Spearman correlation between the induction of the CC-oscillator signatures and clinical demographics and the PLS in 216 early-stage (i.e., Child-Pugh class A) HCV cirrhosis patients[38]. The signatures were defined as genes with >4-fold over-expression in either HCV-infected or control hepatocytes in early or late timepoints in each of the four representative models (i.e., loss, gain, altered, and unaltered). All pair-wise correlations between the variables (correlation matrix) as shown in rows and columns. The correlation matrix was clustered to depict groups of variables sharing similar patterns of correlation. The deep blue, light blue, and brown color bars indicate the presence of the three major

correlation clusters, a two-tailed test. Source data are provided as a Source Data file. **c** Classification of the 216 early-stage cirrhosis patients[38] by the CC-oscillator signatures associated with clinical features related to liver disease severity (Supplementary Table 2), magnitude of gene signature/set modulation is shown in each patient (in each column) in the cohort. The induction/suppression of the circadian clock gene sets (shown in the bottom half of the panel) was determined by GSEI[21]. For the PLS, normalized expression levels or the PLS member genes were used for the NTP analysis[18,38]. AFP alpha-fetoprotein, ALT alanine aminotransferase, AST aspartate aminotransferase, PLS prognostic liver signature, SVR sustained virologic response. Source data are provided as a Source Data file. **d** Association of the disease severity-related HCV$_{CLOCK}$ signature-based classification with overall survival in the 216 early-stage HCV cirrhosis patients[38]. $p = 0.03$, log-rank test, two-tailed. Source data are provided as a Source Data file. **e** The Hallmarks of Cancer and perturbed rhythmic pathways in the humanized liver (HCLM), which were altered by HCV infection, predisposing toward HCC.

reverse transcriptase (ReverTra Ace; TOYOBO Co., LTD., Osaka, Japan) according to the instructions provided by the manufacturer. HCV RNA levels in mice were measured using the COBAS TaqMan HCV test (Roche Diagnostics, Tokyo, Japan), using the primers- KY80M: 5′-TTCACGCAGAAAGCGTCTAGC-3′, and KY78M: 5′-GCAAGCACCCTAT CAGGCAGT-3′. The results are expressed as international units (IU)/mL. The lower detection limit for the assay in mice was 3.45 log IU/mL.

### RNA-seq NGS library preparation
The amount of total RNA was quantified using the Qubit 2.0 Fluorometric Quantitation system (Thermo Fisher Scientific, Waltham, MA, USA), and the RNA integrity number (RIN) was determined using the Experion Automated Electrophoresis System (Bio-Rad, Hercules, CA, USA). RNA-seq libraries were prepared with the TruSeq Stranded mRNA LT sample preparation kit (Illumina, San Diego, CA, USA) using Sciclone and Zephyr liquid handling workstations (PerkinElmer, Waltham, MA, USA) for pre- and post-PCR steps, respectively. Library concentrations were quantified with the Qubit 2.0 Fluorometric Quantitation system (Life Technologies, Carlsbad, CA, USA), and the size distribution was assessed using the Experion Automated Electrophoresis System (Bio-Rad, Hercules, CA, USA). For sequencing, samples were diluted and pooled into NGS libraries in equimolar amounts. Expression profiling libraries were sequenced on HiSeq 3000/4000 instruments (Illumina, San Diego, CA, USA) following a 50 bp, single-end recipe. Raw data acquisition (HiSeq Control Software, HCS, HD 3.4.0.38) and base calling (Real-Time Analysis Software, RTA, 2.7.7) was performed on-instrument, while the subsequent raw data processing of the instruments involved two custom programs based on Picard tools (2.19.2). In the first step, base calls were converted into lane-specific, multiplexed, unaligned BAM files suitable for long-term archival (IlluminaBasecallsToMultiplexSam, 2.19.2-CeMM). In a second step, archive BAM files were demultiplexed into sample-specific, unaligned BAM files (IlluminaSamDemux, 2.19.2-CeMM).

### H3K27ac and H3K9ac chromatin immunoprecipitation (ChIP)
Chromatin immunoprecipitation (ChIP) of H3K27ac and H3K9ac were performed as described below[35]. In brief, for ChIP from humanized mice, we used ~100 mg liver per timepoint. Liver tissues were dounce homogenized in cold PBS, on ice. Subsequently, fresh formaldehyde was added (1% final concentration (v/v) to the PBS-containing cells and incubated on a flip-flop rocker for 10 min at room temperature, followed by the addition of 2 M glycine (0.125 M final concentration) and incubation for 5 min (room temperature). Cross-linked cells were pelleted (400×*g*, 5 min) at 4 °C, washed twice in ice-cold PBS and resuspended in ice-cold lysis buffer [10 mM EDTA, 50 mM Tris-HCl (pH 8), 1% SDS, 0.5 x PMSF and protease inhibitor cocktail Complete, EDTA-free (Roche)]. Cross-linked lysates were sonicated [Covaris; (Power 200 Watts, Duty factor 28%, Cycles/burst 200 and time/sample

20 min)] at 4 °C to generate 200–500 bp chromatin fragments. Cellular debris were removed by centrifugation (10,000×*g*, 10 min) at 4 °C, and the supernatant (chromatin extract) was quantified and verified for sonication efficiency. Next, the chromatin extracts were pre-cleared with 30 μL of protein G Dynabeads (per sample) for 1 h at 4 °C. Next, beads were discarded, and equal amounts of lysates were subjected to immunoprecipitation. 10% of the lysate was saved and served as input DNA. Next, individual samples were diluted 1:8 (v/v) in ChIP dilution buffer (16.7 mM Tris, 0.01% SDS, 1% Triton X-100, 1 mM EDTA, 16 mM NaCl, 1X Complete, EDTA-free (pH 8.1) and incubated overnight at 4 °C on a rotating wheel with control IgG (Diagenode, #C15410206; 1 μg/sample) and either H3K27ac (Active Motif, #39133; 5 μg/sample) or H3K9ac antibody (Active Motif, #39137; 5 μg/sample). Next, 50 μL of pre-cleaned Dynabeads G were added for 60 min at 4 °C on a rotating wheel. Protein G-bound immunocomplexes were then washed twice with low-salt buffer (20 mM Tris, 0.1% SDS, 1% Triton X-100, 2 mM EDTA, 150 mM NaCl, pH 8.1), twice in high-salt buffer (20 mM Tris, 0.1% SDS, 1% Triton X-100, 2 mM EDTA, 500 mM NaCl, pH 8.1), twice in LiCl buffer (10 mM Tris, 250 mM LiCl, 1% NP-40, 1% sodium deoxycholate, 1 mM EDTA, pH 8.1), and finally washed twice in 1 mL TE buffer (10 mM Tris, 1 mM EDTA, pH 8). Chromatin was released from the beads by incubation with 250 μL of elution buffer (1% SDS, 100 mM NaHCO₃, pH 8) for 10 min at room temperature on a flip-flop rocker. The elution step was repeated, and both eluates were pooled. 1 μL of RNaseA (10 mg/mL; Fermentas) and NaCl (200 mM final concentration) was added to the eluate and incubated at 65 °C overnight. The reaction was stopped by 3 μL proteinase K (10 mg/mL; Fermentas) and incubated at 50 °C for 1 h. DNA was subsequently purified from the eluate using a QIAGEN PCR purification kit in a final volume of 50 μL. This DNA was used for library preparation.

### ChIP-seq NGS library preparation
ChIP samples were purified using Agencourt AMPure XP beads (Beckman Coulter) and quantified with the Qubit (Invitrogen). ChIP-seq libraries were prepared from 3 to 10 ng of double-stranded purified DNA using the MicroPlex Library Preparation kit v2 (C05010014, Diagenode s.a., Seraing, Belgium), according to manufacturer's instructions. In the first step, the DNA was repaired and yielded molecules with blunt ends. In the next step, stem-loop adapters with blocked 5-prime ends were ligated to the 5-prime end of the genomic DNA, leaving a nick at the 3-prime end. The adapters cannot ligate to each other and do not have single-strand tails, avoiding non-specific backgrounds. In the final step, the 3-prime ends of the genomic DNA were extended to complete library synthesis, and Illumina-compatible indexes were added through a PCR amplification (7 + 4 cycles). Amplified libraries were purified and size-selected using Agencourt AMPure XP beads (Beckman Coulter) to remove unincorporated primers and other reagents. Libraries were sequenced on an Illumina HiSeq 4000 sequencer as single read 50 base

reads. Image analysis and base calling were performed using RTA version 2.7.7 and bcl2fastq version 2.20.0.422.

## Patient cohorts

The chronically HCV-infected patients (F1–F4) and control subjects without liver disease have been described[17,35]. The MASH cohort has been described[19]. For the study described by ref. 17 all patients provided written consent, and the study was approved by the ethics commission of the cantons Basel-Stadt and Basel-Land (approval number EKBBM189/99). For the studies described in Hamdane et al.[35] and Jühling et al.[19], samples were obtained with informed consent from all patients for de-identified use, and protocols were approved by the ethics committee of the Strasbourg University Hospital (DC-2016-2616), Mount Sinai Hospital, New York (HS13-00159), and Basel University Hospital, Switzerland (EKNZ2014-362). For the 216 early-stage (Child-Pugh class A) cirrhosis patients described by Hoshida et al.[38], the study was approved by the review board of each participating institution on the condition that all samples were anonymized. The three HCV-associated HCC DAA-cured paired liver tissues were obtained from patients followed at the Gastroenterology and Hepatology Clinic of the Hiroshima University Hospital (Hiroshima, Japan), and the study protocol for human patient samples was approved by the Hiroshima University Hospital ethical committee (approval number HI-98-21) in accordance with the Helsinki declaration. All patients provided written informed consent. Most of the liver specimens in these cohorts were obtained in the morning, however, we cannot exclude a moderate temporal heterogeneity due to variation in clinical schedule.

## Transcriptome analyses

NGS reads from humanized mouse livers and human patient samples were mapped to an artificial genome containing the human and mouse Genome Reference Consortium GRCh38 and GRCm38 assemblies (humanized mouse) or the common GRCh38 assembly using HISAT2[55]. Mouse and human reads were counted with htseq-count[56], and normalization as well as differential expression analysis performed with Bioconductor DESeq2 (1.28.1)[57] package based on a model using the negative binomial distribution. Clustering for selecting the closest samples at each ZT was performed using ward.D2 algorithms as implemented in the Bioconductor ComplexHeatmap package[58].

## ChIP-seq analyses

Raw reads were demultiplexed using bcl2fastq v2.20.0.422 mapped to the human genome hg38 using the Bowtie aligner v1.2.2[59]. Quality control checks were performed using FastQC. Bigwig files were created using make UCSCfile from HOMER[60] with the parameters- norm 14028944-fraglength 200. All original alignment files were scaled down to the same number of reads set as the lowest number of the mapped reads (14 M for H3K27ac) or to the maximum of 20 M (in the case of H3K9ac) using the tool MACS2 randsample[61]. with the following parameters-n14028944-f BAM. All original Peaks were then called in uniquely mapped reads using MACS3[62] v3.0.0b3 with parameters -f BED-g hs. Target genes were defined using the predicted transcription factor binding sites listed in the JASPAR[63] database with a score higher than 500.

## The *dryR* analyses of cycling genes and pathways

Differences in cycling gene pattern and assignments to expression modules comparing mouse vs. human hepatocytes, as well as between control (uninfected human hepatocytes) and HCV-infected human hepatocytes, were analyzed using *dryR*[25]. In the case of comparing rhythmicity between human hepatocytes and mouse hepatocytes, only genes with orthologues in both species and with a mean of more than 10 read counts were used as input for the *dryR* analyses[26]. Intersections with external data were performed based on human gene names and hypergeometric tests were calculated for overlaps. Gene set enrichment

for MsigDB v7.2[64] sets comparing conditions were calculated using local javaGSEA, and sample-wise enrichments were calculated using GSVA[65]. Genome tracks were generated using the Bioconductor karyoploteR[66] package based on down-samples BAM alignment files, heatmaps were generated using the Bioconductor ComplexHeatmap[58] package, and other customized plots were generated using ggplot2.

## Assessment of $HCV_{CLOCK}$ signature

To assess the presence of the 4 CC-oscillator transcriptomic model (i.e., loss, gain, altered, and unaltered) in liver disease patient cohort for association with clinical characteristics and prognosis, CC-oscillator gene sets were first defined as differentially expressed genes (at fourfold or more) between early (ZT0-ZT8) or late (ZT12-ZT20) timepoints in each of HCV-infected and control conditions in each model. Subsequently, enrichment of the gene sets were assessed in our previously generated genome-wide transcriptome profiles of 216 early-stage (Child-Pugh class A) HCV-related cirrhosis patients[18] (GSE 15654) by using the esearch algorithm utilizing Kolmogorov–Smirnov statistic-based enrichment score[21,47] implemented in the esearch module of GenePattern genomic analysis toolkit[67] (github.com/genepattern/gparc-module-docs). Gene set enrichment index (GSEI) was calculated as −log10 (nominal $p$ value for the enrichment score) with a sign of enrichment score for each signature and each patient[21,47]. GSEI was also calculated for the PLS that predicts poor prognosis of liver cirrhosis patients[18,21,38,44–47]. The correlation of GSEIs with clinical characteristics and PLS was evaluated by the Spearman rank correlation test. Using the CC-oscillator signatures correlated with clinical features reflecting liver disease severity, the 216 cirrhosis patients were classified into high- or low-risk groups using the Nearest Template Prediction (NTP) algorithm[68] implemented in the Gene Pattern Nearest Template Prediction module. For the $HCV_{CLOCK}$ risk prediction, the GSEIs of the circadian clock gene sets for the NTP analysis were used. Association of the classification with time to overall death was assessed by Kaplan–Meier method and Cox regression. All analyses were performed using R statistical language (www.R-project.org).

## Histology and immunohistochemistry

Samples for histology were placed in 10% neutral formalin overnight before transfer to 70% ethanol and later embedding in paraffin and cross-sectioned to obtain a 5 μm section, and then extra-coated with paraffin to preserve tissue integrity. Glass slide-mounted tissues were scanned with Nanozoomer scanner (Hamamatsu), and images were analyzed using image processing software (ImageJ). Additional sections were stained with antibody specific for cMYC, and CK18. Deparaffinization was performed in the BondMax automate (Leica biosystem) in Bond Dewax Solution or in a xylene bath and decreasing alcohol (for CK18). Antigen unmasking was performed in the BondMax in EDTA buffer plus detergent (pH 8.9–9.1 at 95 °C), or 10 mM citrate buffer (pH 6; CK18). Sections were preincubated with 3% $H_2O_2$ for 5–10 min, incubated for 20 min in blocking buffer (ABC kit, Vector Labs), and then incubated with primary antibodies (CK18, M701029-2, Clone DC 10, Dako; 1:15; cMYC, #ab32072 Active Motif 1:100) according to manufacturer's instructions. After subsequent washes with Bond Wash Solution (Leica), sections were incubated with the secondary antibody for 20 min. After washing, sections were incubated in Mixed-DAB-Solution for 6 min. Images were taken with a Leica Kit- DS9800 microscope (Leica Biosystems) or NanoZoomer Scan (Hamamatzu) using Plan 20x objectives and analyzed using ImageJ. For H&E and Sirius Red, staining sections were deparaffinized, rehydrated, and processed. For Sirius Red staining: slides were incubated with a 0.1% Sirius Red solution dissolved in aqueous saturated picric acid for 1 h (Sirius Red: Sigma-Aldrich, Direct Red 80, #365548), washed in acidified water (0.5% HCl), dehydrated, and mounted under coverslips with Eukitt mounting medium (Sigma). For HE staining, the deparaffinized and rehydrated sections were washed up and stained with Mayer's

Hematoxylin and eosin (Sigma) solution. Briefly, whole slide images were generated using 3D Panoramic SCAN or Nanozoomer Scan and uploaded into Ndpi view software. For collagen-positive area (CPA) morphometric quantification was performed in the Sirius Red -stained slides and quantified as a percent of the total image analysis area using ImageJ (NIH). Quantification of CPA and HE slides to determine fibrosis and steatosis, respectively were done blindly.

### Antibodies

ChIP was conducted using antibodies targeting H3K27ac (Active Motif; #39133), H3K9ac (Active Motif, #39137) and IgG (Diagenode, #C15410206). Following antibodies were used for IHC: CK18 (CK18, #M701029-2, Clone DC 10, Dako), and MYC (Abcam, #ab32072). All antibodies were validated by the vendor for their specificity and target detection.

### Statistical analysis

Gene expression and pathway enrichments were compared on bulk level for each ZT separately. Gene expressions were compared using DESeq2 and then using the Wald test ($p < 0.05$). Gene set enrichment scores were compared by applying the Wilcoxon signed-rank test ($p < 0.05$) as implemented in R on enrichment scores.

### Reporting summary

Further information on research design is available in the Nature Portfolio Reporting Summary linked to this article.

## Data availability

The RNA-seq and ChIP-seq data generated in this study have been deposited in the National Center for Biotechnology Information's Gene Expression Omnibus (GEO) under accession code GSE200812. Previously published data used in this study are available in the GEO database under accession codes GSE84346 and GSE15654, and in the SRA database under accession codes SRP170244 and SRP248410. Source data are provided with this paper.

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

## Acknowledgements

We acknowledge the Center de Ressources Biologiques (Biological Resource Center), Strasbourg, France, for the management of patient-derived liver tissues. We also thank the GenomEast platform (IGBMC, Illkirch, France), a member of the "France Genomique" consortium (ANR-10-INBS-0009) for library preparation and sequencing of ChIP samples. We thank D. Heide (DKFZ, Heidelberg) for assistance with MYC immunostaining. We thank Dr. Mirjam B. Zeisel, Dr. Carla Eller, and Dr. Katharina Herzog for their assistance in the initial phase of the study. We thank Dr. Hideki Ohdan and his surgical group in the Department of Gastroenterological and Transplant Surgery, Hiroshima University who helped to store and use liver specimens obtained in the surgery. The authors acknowledge funding by the European Union (ERC-AdG-2020-FIBCAN # 101021417 to T.F.B. and Y.H., HORIZON-HLTH-2021-DISEASE-04-07 D-SOLVE #101057917 to T.F.B. and J.L.), Fondation ARC (Ther-aHCC2.0 IHUC201901299 to T.F.B.), the University of Strasbourg Institute for Advanced Study (USIAS-2020-029 to A.M.), the University of Strasbourg Foundation (T.F.B. and C.S.), the Alsace Cancer Foundation (T.F.B.), the ANRS Maladies infectieuses émergentes (ANRS-MIE; ECTZ103701 to T.F.B., ECTZ160436, ECTZ131760, to J.L.), the French National Research Agency RHU DELIVER (ANR-21-RHUS-0001) and LABEX ANR-10-LABX-0028_HEPSYS (T.F.B.), the US National Institute of Health (U19-AI123862) to T.F.B and S.K.H.F., and the US National Institute of Health (R01CA233794) to Y.H. and T.F.B., the UK Medical Research Council (MRC; MR/R022011/1 to J.A.M.), the Wellcome Trust (IA 200838/Z/16/Z to J.A.M.) and the Japan Agency for Medical Research and Development (AMED; 19fk0210020h0003 to K.C.), and the Inserm Plan Cancer 2019–2023 to T.F.B. Y.H. acknowledges the funding from US National Institute of Health (CA233794, CA255621), and Cancer Prevention and Research Institute of Texas, USA (RR180016, RP200554). The work in the lab of M-L. Y was partly supported by the "Center of Excellence for Metabolic Associated Fatty Liver Disease, National Sun Yat-sen University, Kaohsiung" from the featured areas research center program within the framework of the Higher Education Sprout Project by

the Ministry of Education (MOE), Taiwan. This work of the Inter-disciplinary Thematic Institute IMCBio, as part of the ITI 2021–2028 program of the University of Strasbourg, CNRS, and Inserm, was supported by IdEx Unistra (ANR-10-IDEX-0002) and by the SFRI-STRATUS project (ANR 20-SFRI-0012) and EUR IMCBio (ANR-17-EURE-0023) under the framework of the French Investments for the Future Program.

## Author contributions

T.F.B. initiated and coordinated the study. J.L. and T.F.B. jointly supervised the study. A.M., J.L., and T.F.B. designed experiments. F.J. performed the computational analyses and representation of NGS data and figures. A.M., Y.S., E.C., F.D.Z., Y.T., A.H., P.B., C.G., N.S., M.A.O., S.C.D., C.P., X.Z., and H.A.C. performed experiments. A.S., M.I., M.C., Y.H., and K.C. provided samples from patient cohorts. K.C. provided liver tissue samples from HLCMs. A.S., M.I., M.C., and Y.H. generated the HCV$_{CLOCK}$ gene signature. S.P., F.R., N.F., and Y.H. performed the bioinformatic analyses of cancer gene and HCV$_{CLOCK}$ signatures. A.M., F.J., J.L., and T.F.B. wrote the manuscript. M.H., J.H., R.T.C, M.L.Yeh, M.L.Yu, I.D., S.K.H.F., C.S., and J.A.M. edited and revised the manuscript. All the authors read and approved the manuscript contents and the author list and its order.

## Competing interests

The authors declare no competing interests.

## Additional information

[1]University of Strasbourg, Institute of Translational Medicine and Liver Diseases (ITM), Inserm UMR_S1110, Strasbourg, France. [2]Department of Gastroenterology, National Hospital Organization Kure Medical Center, Hiroshima, Japan. [3]Department of Functional Genomics and Cancer, Institut de Génétique et de Biologie Moléculaire et Cellulaire (IGBMC), CNRS/INSERM/University of Strasbourg, Illkirch, France. [4]Department of Internal Medicine, University of Texas Southwestern Medical Center, Dallas, TX, USA. [5]Nuffield Department of Medicine, University of Oxford, Oxford OX3 7FZ, UK. [6]Institute of Immunity & Transplantation, Division of Infection & Immunity, UCL, Pears Building, Rowland Hill St, London NW3 2PP, UK. [7]University of Melbourne, St Vincent's Hospital, Melbourne, VIC, Australia. [8]Hepatobiliary Division, Department of Internal Medicine, School of Medicine and Hepatitis Research Center, College of Medicine, and Center for Liquid Biopsy and Cohort Research, Kaohsiung Medical University Hospital, Kaohsiung Medical University, Kaohsiung 80708, Taiwan. [9]Center for Medical Specialist Graduate Education and Research, Hiroshima University, Hiroshima, Japan. [10]Division of Chronic Inflammation and Cancer, German Cancer Research Center (DKFZ), Heidelberg, Germany. [11]M3 Research Center, Tübingen, Germany and Cluster of Excellence iFIT (EXC 2180) "Image-Guided and Functionally Instructed Tumor Therapies, " Eberhard-Karls University of Tübingen, Tübingen, Germany. [12]Division of Gastroenterology and Hepatology, Fondazione IRCCS Cà Granda Ospedale Maggiore Policlinico, Milan, Italy. [13]EASL International Liver Foundation, Geneva, Switzerland. [14]Department of Pathology, Stanford University School of Medicine, Stanford, CA 94305, USA. [15]Chinese Academy of Medical Sciences Oxford Institute, University of Oxford, Oxford, UK. [16]School of Medicine and Doctoral Program of Clinical and Experimental Medicine, College of Medicine and Center of Excellence for Metabolic Associated Fatty Liver Disease, National Sun Yat-sen University, Kaohsiung, Taiwan. [17]Gastrointestinal Division, Hepatology and Liver Center, Massachusetts General Hospital, Boston, MA 02114, USA. [18]RIKEN Center for Integrative Medical Sciences, Yokohama, Japan. [19]Hiroshima Institute of Life Sciences, Hiroshima, Japan. [20]Gastroenterology and Hepatology Service, Strasbourg University Hospitals, Strasbourg, France. [21]Institut Universitaire de France, Paris, France. [22]IHU, Strasbourg, France. [23]These authors contributed equally: Atish Mukherji, Frank Jühling. [24]These authors jointly supervised this work: Joachim Lupberger, Thomas F. Baumert. ✉e-mail: joachim.lupberger@unistra.fr; thomas.baumert@unistra.fr

