## [Peer Review File · Nature Communications]

An atlas of the human liver diurnal transcriptome and its perturbation by hepatitis C virus infectionEditorial Note: This manuscript has been previously reviewed at another journal that is not operating a transparent peer review scheme. This document only contains reviewer comments and rebuttal letters for versions considered at Nature Communications.

Reviewers' Comments:

Reviewer #4:

Remarks to the Author:

The authors have significantly improve the manuscript including in vitro validations of several claims and increasing the sample size to an acceptable number.

Major Concerns:

It is unclear to me how the 2-replicates from the first experiment were chosen in the first half of the paper, why weren't all 3 of the original experiment included?

Minor Concerns:

Extended Data Fig 4 -> the authors should perform a hypergeometric test (or Fisher's exact test) to determine the significance of this overlap. From my back of the envelop calculation it seems it is likely these are not significant.

Extended Data Fig 5 -> the authors should double check their classification as "similar pathways" and "distinct pathways" because several "similar pathways" appear to be anti-correlated between human and mouse e.g. KE_VIBRIO_CHOLERAЕ_INFECTON

"Furthermore, we found that several of the zinc 9 finger TFs (ZNFs) were solely circadian in human hepatocytes (Extended Data Fig. 6)." - Wouldn't it be more accurate to say several of the ZNFs were only expressed in human hepatocytes as Ext.Fig 6 has grey squares representing no expression in the mouse data. How many of these ZNFs even have mouse orthologs? Are these simply human-specific genes?

Figure 2 - This whole figure is lacking information on statistical significance. Fig 2b should include a "*" for which Log2FCs were significant and Figure 2D would benefit from a marker (e.g. a star) indicating which time points or which pathways are statistically significantly different and which are not.

Reviewer #5:

Remarks to the Author:

Below you will find in blue my opinion on the authors' response to reviewer 1.

Please find in blue below my opinion on the authors' response to reviewer 1.

MS#NMICROBIOL-22051240-T

POINT-BY-POINT RESPONSE TO THE REVIEWERS

Reviewers 1 and 2: Summary

In this manuscript Mukherji et al use a human liver chimeric mouse model to observe the diurnal transcriptome and epigenome in human hepatocytes and investigate how hepatitis C virus (HCV) infection alters this. Their study generated human liver chimeric mice, infected them with HCV and harvested mice at various zeitgeber timepoints for bulk RNA-seq and ChIP-seq of human and mouse hepatocytes. Their main findings from this experiment are; 1). That engrafted human hepatocytes (confirmed to have active circadian cycle oscillating genes) display many differences compared to mouse hepatocytes in terms of genes, transcription factors and associated pathways showing rhythmic expression and that diurnal variations of enhancer and promoter associated H3K27ac followed transcriptional changes seen. 2). HCV perturbs the circadian transcriptome causing loss of rhythmicity in core circadian cycle oscillator genes as well as in various genes and pathways potentially relevant for disease processes such as fibrosis, steatosis and hepatocarcinogenesis. 3). HCV disrupts H3K27ac levels in promoter-enhancer regions of cycling genes causing loss of circadian variation and hyper-H3K27 acetylation. On further analysis HCV also affected H3K27ac levels around various HCV host factors as well as known regulators of histone acetylation such as Sirtuins. The authors propose this effect of HCV on Sirtuins (such as SIRT1) as being the possible mechanism behind HCV-induced CC-rhythmic variation and that the resulting loss of CC-rhythmic variation in various pathways helps drive progression of liver disease and hepatocarcinogenesis. 4). Finally the authors attempt to compare and contrast these findings to various patient liver biopsy cohorts. Showing differential gene expression levels of various circadian cycle oscillator genes and pathogenic pathways which do not fully resolve after sustained virologic response (SVR). The authors then used these data to generate a predicative gene signature in a 5th patient cohort showing that it correlated with the prognostic liver score (PLS), various clinical markers of disease severity and survival.

I applaud the authors in their novel approach at studying the effect of HCV on hepatocyte circadian cycle. Previous studies have used either mice or human cell lines in vitro. Human liver chimeric mouse models allow for an in vivo approach but with human hepatocytes, allowing observations of circadian cycle effects in a context closer to those of in situ human hepatocytes. The clinical context of risk of progression of liver disease and development of HCC is also important and a major challenge in the HCV field despite effective viral clearing drugs. However this study has several major conceptual and experimental flaws which make it unsuitable for publication in its current state (see detailed major and minor comments). The main flaws are; 1). The study is purely observational and based on multiple analysis of one experiment. The authors need to validate their key findings by alternate measures and prove their mechanism in further experiments. Multiple analysis of the same dataset is not enough 2). The robustness of the data from the human liver chimeric mouse experiment is questionable with only two biological repeats (n=2). I suspect many of the trends the authors highlight are not statistically significant as a result. 3). There are a number of problems with the HCV patient cohort data. The comparisons of HCV with fibrosis vs healthy liver samples are confounded by fibrosis/chronic liver disease. The differential gene expression may not be HCV specific and instead be a result of fibrosis/chronic liver disease. Comparisons between RNA isolated from whole patient biopsies vs human hepatocytes in chimeric mouse model are confounded by the different cell populations included in the biopsies (immune cells, cholangiocytes etc). 4). The focus of this work is on the effects of HCV on hepatocytes and potential liver disease

progression. No further work on manipulating the viral factors has been done to investigate further mechanism.

Answer: We sincerely thank the reviewers for their thorough evaluation of our study and their general appreciation of the manuscript “**applaud the authors in their novel approach at studying the effect of HCV on hepatocyte circadian cycle. Previous studies have used either mice or human cell lines in vitro. Human liver chimeric mouse models allow for an in vivo approach but with human hepatocytes, allowing observations of circadian cycle effects in a context closer to those of in situ human hepatocytes. The clinical context of risk of progression of liver disease and development of HCC is also important and a major challenge in the HCV field despite effective viral clearing drugs**”. In the revised version we have now fully addressed the remaining conceptual and experimental limitations pointed out by the reviewers. In particular, as have been requested by all reviewers: **(1)** we have now validated our results through multiple independent analyses *in vivo* and *in vitro* and provide a novel mechanism of HCV-induced perturbation of gene expression involving epigenetic modifiers including SIRT1; **(2)** we performed an independent circadian experiment with humanized liver chimeric mice which have now increased the number of biological samples (n=5 chimeric mice/group/circadian timepoint), thus increasing the robustness of our observations; **(3)** we included new patient cohorts and compared the changes induced by HCV and NASH. Moreover, we analyzed the impact of HCV cure in chimeric mice to confirm our findings; and finally **(4)** we revealed the HCV “viral factors” perturbing SIRT1 and BMAL1 expression supporting the mechanistic events leading to the perturbation of the liver clock. A detailed point-by-point response to the individual questions raised by reviewer 1 and 2 is provided below:

Major comments.

1. *This is a descriptive study based on a single experiment with the authors building their conclusions on multiple complex bioinformatic analyses of transcriptomics data. For key pathways and disease processes highlighted I would like further validation. For example, a) confirmation at the protein level either using western blots or proteomics and confirmation of changes in downstream pathways. b) Key associated processes (steatosis and fibrosis) could also be confirmed histologically. c) For ChIP-seq data I would like to see validation using at least one additional marker of histone acetylation. d) For the authors proposed mechanism behind HCV induced alteration of circadian pathways I would like to see further proof experiments e.g. rescue by manipulation of SIRT1 or manipulation of the HCV virus to decipher mechanism. e) I would also like to see validation of the finding of incomplete reversal of HCV associated changes in CCcontrolled pathways. Could this be done in the human liver chimeric mouse model by treatment with anti virals?*

Answer: As suggested, we addressed these points raised in the revised manuscript as follows:

a. To evaluate key pathways and disease processes altered following chronic HCV-infection of humanized liver chimeric mice (HLCM) at the protein level we validated oncogene **MYC** (driver of chronic liver disease), and **SIRT1** (regulator of CC and metabolic pathways). Consistent with our transcriptomic analyses, the immunohistochemistry detected very low levels of MYC protein in the control liver, but its level was increased in the HCV-infected livers (**Extended Data Fig. 15**). Our analyses are consistent with studies noting minimal MYC expressed in control liver and its increased levels only in diseased state (Meyer and Penn, Nat Rev Cancer 2008, 976-990, and Buendia et al., Gastroenterology 2012, 214-218). Furthermore, we performed western blot analysis which showed circadian expression of the SIRT1 in control liver and its reduction in HCVinfected livers (**Extended Data Fig. 19b, c**).

In my opinion, the reviewer's concerns have been resolved. Authors have provided western blot to confirm changes in downstream pathways in HLCM.

b. It is established that due to a lack of adaptive immune cells (T- and B-cells), in human liver chimeric mice, HCV infection leads only to mild fibrosis without significant steatosis (Mercer et al., Nat Med 2001, 927-933, Bissig et al., J Clin Invest 2010, 924-930, and Washburn et al., Gastroenterology 2011, 134-1344). Consistently, our histological analyses **revealed mild fibrosis without any detectable steatosis (Extended Data Fig. 16a-c).**

Authors have provided CK8-18 staining, but

In HCV-infected HLCM liver, no steatosis was detected by histology, while modest but significant liver fibrosis were detected by Sirius red staining at different circadian time points and quantification of collagen positive areas (CPA).

c. As suggested, we have now provided additional ChIP-seq data using a different marker of transcriptional activation: Histone 3 lysine 9 acetylation (**H3K9ac**), which **consolidates our previous findings using H3K27ac on the HCV-induced epigenetic deregulation (Fig. 3c, d, Extended Data Fig. 17b and Supplementary Tables S3, S5).** **The new results are indicated (page 12, lines 21-23, page 13, lines: 1-2, 5-7 and 8-15).**

Authors included an additional marker of histone acetylation, the H3K9ac. They profiled how the HCV infection modifies both, the H3K9ac and H3K27ac in liver from infected and non-infected HLCM.

However, in my opinion authors should also comment these new results in page 9, within the "Circadian regulation of transcription factors and enhancer regions impact pathways relevant for liver homeostasis". Here authors comment that initially they focused on H3K27ac, but they did not comment anything about the new data about H3K9ac, which in my opinion should be addressed in page 9.

d. We have now performed extensive mechanistic studies in well-established cell culture models of HCV replication and infection to address the role of HCV proteins in perturbed expression of the SIRT1 and key CC-gene BMAL1. These include CRISPR-Cas9 guided KO of SIRT1, pharmacological studies using SIRT1 agonist, and luciferase reporter assays. First, **we first confirmed the key role of SIRT1 in the regulation of BMAL1. Secondly, we identified the underlying molecular mechanism of HCV non-structural proteins suppressing SIRT1 expression by inducing PPAR γ (Fig. 4).** These new results are now discussed (**page 14, lines 19-23, page 15, lines 1-23, and page 16, lines 1-4).**

Authors have provided new mechanistic experiments regarding the SIRT1 and key CC-gene BMAL1.

e. To address the question of incomplete reversal of HCV associated changes in CC-controlled pathways in human liver chimeric mice treated with antivirals, we analyzed our previously published datasets (Hamdane et al., Gastroenterology 2019, 2313-2329). **Our new analyses confirmed that the HCV-perturbed expression of CC-genes and key CC-controlled pathways driving chronic liver disease and HCC in the livers of humanized liver chimeric mice remain altered following viral cure (Fig. 5a, b; cohort b).**

Authors have used a previously published results (ChIPmentation-based ChIP-Seq on liver tissue with H3K27ac antibody) related to 8 patients with chronic HCV infection cured by DAA treatment and apparently 5 treated with DAAs mice. In my opinion this information should be clarified as according to results showed in Fig. 5a, pooled data of HCV-infected and -cured humanized mice are showed, as well as pooled data of c, d and f patient cohorts. Have the authors provided any statistical analysis to show that

there are no significant differences regarding HCV-perturbed expression of CC-genes following viral cure with DAAs?. These results should be shown. Additionally, information about what is Pat1, Pat 2 and Pat3 should be indicated in figure legend 5.

2. The robustness of the data from the human liver chimeric mouse experiment is questionable with only two biological repeats (n=2). I would like to see ideally n=5, but at the very least an n=3. The initial experiment had an n=3, can the authors do the analysis on this without selecting the two closest samples? I note the error bars on most the graphs are large. I would also like some statistical tests for significance in the differences highlighted by the authors.

Answer: We have now performed an additional set of animal experiments with n=3 mice/group/ circadian time point. This new experiment has been used in the revised manuscript and key results from both the experiments are congruent (**Fig. 1, 2 and Extended Data Fig. 1-19**).

Importantly, **as suggested by the reviewer in this new experiment, we did not “select” samples (Extended Data Fig. S2 and S9). With the addition of this new batch of animals we increased the robustness of our data, and the gene expression curves represent SD of 5 animals (5 chimeric mice/group/ circadian timepoint).** For further pathway analyses **we chose only those circadian pathways which showed FDR <0.05.** Animal numbers and analyses are now described in the revised figure legends.

Here the authors have increased up to 3 the number of animals to increase robustness. However, they do not show any statistical results of DESeq2 to demonstrate differences commented by the reviewer.

3. There are several problems with the patient biopsy cohorts. The controls for cohorts b and d are subjects without liver disease. The changes described may not be due to HCV but due to fibrosis/ chronic liver disease. Indeed there is evidence (in mice) that CCL4 induced fibrosis is associated with altered circadian rhythm of clock genes (doi: 10.1016/j.febslet.2010.03.019). The appropriate comparison would be patients with fibrosis/chronic liver disease from a non-viral etiology (e.g. NAFLD). I would also like more information on how RNA was obtained from the patient biopsy samples. Was RNA extracted from whole tissue or were hepatocytes isolated first?

If the RNA was extracted from whole tissue then the additional cell types (immune cells, cholangiocytes etc) which are not in the human liver chimeric mouse analysis are a confounder when making the comparison.

Answer: As suggested, we have now added transcriptomic data of NASH patients which exhibit alterations of CC genes and CC-pathways expression (**Fig. 5a, b; cohort f**). We would like to emphasize that a recent study (Jouffe et al.; PNAS 2022, Vol119, 10.e2200083119) noted perturbed CC-genes expression in NAFLD patients without fibrosis. Furthermore, our recent studies (J Hepatol-S-23-01054-R1, and J Hepatol 78, S77-78) also found CC-perturbation in NAFLD/NASH patients. Thus, **CC-disruption seems to be an integral part of liver disease of different etiologies. We have now modified the text accordingly (page 18, lines 2-10).** As the reviewer can appreciate that most of the samples are **‘fine needle’** biopsies and from infectious origin with barely enough materials making hepatocyte isolation technically challenging, **we utilized whole RNA for these experiments. However, our chimeric mice studies do confirm an effect of HCV-infection on human hepatocytes.**

Transcriptomic data of NASH patients have been included. However, they refer to heatmap figures to show differences between group of patients, but do not show statistical test (DESeq2 results) to confirm the visual trends observed in heatmaps.

Additionally, according to authors "Our analyses indicate that compared to healthy controls both HCV 6 and NASH patients display perturbations of CC gene expression (Fig. 5a, cohorts c and f).(lines 5-6 page 18)", however this figure do not show statistical differences of these comparisons for each gene displaying perturbations of expression. Regarding how the RNA was obtained from the patient biopsy samples, authors have resolved the reviewer concern about the RNA isolation procedure. However, they have not included this clarification in the text. This information should be clearly stated in material and method (page 34, line 4), as it has been clarified in results section (page 12 lines 1- 15)

Additionally, it should be commented how the different library kit used for HLCM samples (TrueSeq stranded) and human liver samples (NEBNext) could introduced some bias in the in the results.

4. *The manuscript needs more details of the human liver chimeric mouse model. Cryopreserved hepatocytes were obtained from BD biosciences, however if possible some basic clinical details of the source of these cells and their phenotype e.g. patient age, indication ? malignant? Also, information on the handling of these hepatocytes should be included. If they were defrosted and immediately implanted there are potential issues with stress and temperature shock, especially given cold temperatures have profound effects on clock genes (doi: 10.15252/embj.2020105604). If they were in culture for a prolonged period of time this too could be problematic as hepatocytes rapidly dedifferentiate in culture with hepatocyte-determining transcripts being down-regulated within hours (doi: 10.1046/j.1365-2613.1998.00083.x). I note there is a difference in the amount of repopulation of human hepatocytes in the control vs HCV groups (human albumin 14,203,835 vs 9,408,337 $p < 0.0001$, %HS 0.74 vs 0.53 $p < 0.0001$). The authors need to comment on this. I am concerned that this could be due to using a different batch of hepatocytes for control and HCV groups or due to hepatocyte death directly caused by HCV. Both of these would be major confounders. I would also like to see histology from these chimeric livers assessing cell death, tissue architecture and stains to differentiate human vs mouse hepatocytes (e.g. hCK8/18).*

Answer: We have now provided more details on the chimeric mice and the PHH in the methods (page 32, lines 19-22, page 33, lines 1-2). We would like to emphasize that our engraftment procedure is state-of-the-art and identical to a recently published study on human hepatocytic clock in chimeric mice model (Delbés et al., *Sci. Adv.* 2023), thus suggesting this procedure does not lead to 'potential issues with stress and temperature shock' on clock genes.

Authors have provided more information on the human liver chimeric mouse model.

As suggested by the reviewer we have now performed an additional set of experiment with control (n=18) and HCV-infected (n=18) mice, and in our **both sets of experiment the degree of humanization in control was between 65-70% (Table S1 and Extended Data Fig. 1)**. Indeed, as suggested by the reviewer our analyses found differences in serum albumin levels and CK18 staining between control and HCV-infected group (**Extended Data Fig. 10 and Supplement Table S1**). This reduction in human albumin probably reflects more murine hepatocytes trying to repopulate in the infected liver (Extended Data Fig. 10; Mercer et al., *Nat Med* 2001, 927-933, and Bissig et al., *J Clin Invest* 2010, 924-930). However, it should be noted that **our two independent RNA-seq found no significant differences in human-specific total reads between control (n=36; 2 experiment) and HCV-infected mice (n=36; 2 experiment) (Supplementary Table S2).**

Additionally, the referee was concerned about the difference in the amount of repopulation of human hepatocytes in the control vs HCV groups. Instead of commenting on these references, the authors have provided a set of additional experiments with a

higher similarity of humanization level. Authors have confirmed that both, control and HCV-infected mice were engrafted with the same batch of PHH (page 33, line 1). Regarding the two independent RNA-seq analysis, authors comment that no significant differences in human-specific total reads were found between control and HCV-infected mice. However, any statistical test has been provided to prove this. In fact, I performed independent t-test analysis for each RNA-seq series 1 and 2, and significant statistical differences are found between both groups (0.001 and 0.046). To compare total number of reads in the two experiments (n=18 series 1 and n=18 series 2) a previous normalization with all samples should be performed. Additionally, a significant different number of total reads were obtained for each series of RNA-seq, and therefore this different be accounted (different batch) to verify that no significant different were found between controls and HCV-infected mice.

5. *The authors have presented the majority of the data in the main figures as heatmaps. Whilst these are visually pleasing I find them very difficult to interpret separate to the main text. For some even after reading the main text and repeatedly studying the figures I was unable to see the same patterns/ interpretation. I am concerned that general readers will struggle. Examples a). I could not interpret figure 4c, not understanding why each variable appears to be repeated and did not understand the meaning of the brown and blue bars. b). In figure 2c the meaning of the target plots and different colour scheme (green alone, red alone, brown and red and green) is not explained. c). The different colored heatmaps for mouse vs human and HCV vs control make it difficult for the reader to make direct comparisons themselves. I would advise using the same colours as they are clearly labelled. I would also advise the authors displaying all the genes in 1b and 2b in extended figures 2 and 7 respectively as trends more easily seen.*

Answer: As, suggested by the reviewer we have now revised the color schemes of the heatmaps for more clarity.

a) We have now explained the interpretation of previous Fig. 4c in a more lucid manner in the main text and the figure legends. “Importantly, we observed changes in gene set enrichment index (GSEI) which indicates ‘magnitude of correlation’ with overall survival of the patients according to HCVclock classification (Fig. 5d, e), suggesting its possible clinical relevance” (**page 19, lines 1-3**). **Figure legend 5c now reads:** “Correlation between induction of the CC-oscillator signatures and clinical demographics and our previously reported Prognostic Liver Signature (PLS) in 216 early-stage (i.e., Child-Pugh class A) HCV cirrhosis patients. The signatures were defined as genes with > 4-fold over-expression in either HCV-infected or control hepatocytes in early or late time points in each of the 4 representative models (i.e., loss, gain, altered, and unaltered). All pair-wise correlations between the variables (correlation matrix) as shown in rows and columns. The correlation matrix was clustered to depict groups of variables sharing similar patterns of correlation.” **The brown, deep blue and light blue color bars indicate the presence of the 3 major correlation clusters. Figure legend 5d now reads:** “Classification of the early-stage cirrhosis patients by the CC-oscillator signatures associated with clinical featured related to liver disease severity (Supplementary table 7), Magnitude of gene signature/set modulation is shown in each patient (in each column) in the cohort. The induction/suppression of the circadian clock gene sets (shown in bottom half of the panel) was determined by Gene Set Enrichment Index (GSEI) as previously described (Nakagawa et al., Cancer Cell 2016, 879-890). The PLS and HCVclock risk predictions were determined by using Nearest Template Prediction (NTP) algorithm (Hoshida, Plos One 2010, e15543). For the PLS, normalized expression levels or the PLS member genes were used for the NTP analysis as previously described (Hoshida et al., N Engl J Med 2008, 1995-2004, Hoshida et al., Gastroenterology 2013, 1024-1030). For the HCVclock risk prediction, we used the GSEIs of the circadian clock gene sets for the NTP analysis.”

Authors have clarified the figure 5d and e (previously figure 4d)

b) We have revised the text explaining the meaning of the target plots and different color schemes for Fig. 2c (Page 11, lines 11-15).

Figure 2c has been clarified

c) We have now used same-colored schemes for the heat maps mouse vs human (Fig. 1b and 1d). All the genes shown previously in Figs. 1b, 2b are now also present in the revised Extended Figures 3a and 11b.

Reviewer wanted to see same coloured schemes for either mouse vs human comparison and HCV vs control, which has not been addressed:

Figure 1b: blue for human green for mouse.

Figure 1c: blue for series 1 and green for series 2 of human hepatocytes in HLCM.

Figure 1 e: which according to authors shows "Circadian expression pattern of transcription factors (TFs) in human hepatocytes (red) and in 11 residual murine (green) cells in HLCM (n=5 HLCM/ circadian timepoint), as predicted by JTK." However, the image showed 7 collapsed heatmaps for each ZT for series 1 and 2.

Figure 1d: black for human and mouse, and blue to red for pathways.

Authors should use the same colour palette for heatmaps of genes and pathways. Dark blue to yellow are used for genes in figure 1, but blue to red in figure 5. On the other hand, figure 5a and 5 b, used same colour for genes and pathways.

Only descriptive data are showed in Figs. 1b, 2b and Extended Figures 3a and 11b, no statistical differences for each gene is showed.

6. The authors do not explain how (or why) HCV brings about the observed circadian changes or the associated alterations in H3K27 acetylation. Without mechanistic investigation and then interventional work the study does not progress sufficiently beyond simple observation. As examples; is this an effect of a particular viral protein or transcript, is there increased expression of a H3K27 acetyl transferase or a demethylase and can this type of effect be associated with the activities of a particular viral protein?

Answer: Our results indicate that the HCV-induced overall increase in H3K27ac and H3K9ac (Fig. 3) originates from a multilayered approach. An overall increase in histone acetylation likely is the result of either an increase in histone acetylase activity/levels or decreased histone deacetylation activity/levels. Of the known histone acetylases, we found HCV-infection increase the transcript levels of only CREBBP (CBP) (Extended Data Fig. 18). We found that HCV does not significantly affect the level of canonical histone deacetylases (Extended Data Fig. 18). However, HCV affects the transcript levels of multiple histone deacetylases belonging to the Sirtuin family, including SIRT1 (Fig. 3h). Importantly, we found that in control liver the **SIRT1 protein** display circadian rhythmicity with a peak in the rest phase, and HCV-infection leads to substantial reduction in peak SIRT1 expression (Extended Data Fig. 19 b, c). In the revised manuscript we revealed that **HCV-infected cells and chimeric mice liver promote the expression PPAR γ , which is a transcriptional repressor of SIRT1** (Han et al., Nucleic Acid Res 2010, 7458-7471, and Yu et al., J Biol Chem 2005, 13600-13605), which correlates with a reduced **SIRT1 expression (Fig. 4)**. **We demonstrate that HCV proteins NS4 and NS5A promote PPAR γ levels while decreasing levels of the epigenetic modifier SIRT1 (Fig. 4)**. In perturbation studies, we established a **functional link between SIRT1 and BMAL1 expression**. Taken together, we demonstrate that HCV employs an elaborate and multistep strategy to alter the acetylation level of histones on a global scale, in which reduction in the levels and/or activity of the deacetylases (Sirtuins) play a major role. However, we believe that further undiscovered mechanisms (e.g., activity of CREBBP, and deregulated expression of circadian TFs) could also be contributing towards altering the epigenome, which remains beyond the scope of this present study and will be addressed in future investigations.

The new results are shown in new **Fig. 3, 4** and **Extended Data Fig. 18, 19** and discussed in the revised manuscript (**page 14, lines 3-23, page 15, lines 1-23, page 16, lines 1-4, page 21, lines 4-22**).

Authors show different observations on transcript levels regarding HCV infection to explain how HCV brings about the commented alterations in H3K27 acetylation. However, first of all these data are only showed as normalized counts and no statistical results from DESeq2 are shown (only bars representing SD). Most of the bars overlap in Extended Data Fig. 18, so therefore no significant different should be present. However, Fig. 3h shows also overlapping in all the ZT for SIRT1, but according to authors HCV affects the transcripts levels of SIRT1, this seems contradictory. If authors provide statistical test results should be easier to understand.

Nevertheless, authors provide an additional set of experiments to demonstrate that HCV proteins decrease levels of SIRT1 (Figure 4).

Minor comments.

1. *Some abbreviations such as PPH and TSS have not been defined and will not be understood by a general reader.*

Answer: We have now explained the abbreviations primary human hepatocytes (PHH; page 6, line 6), and transcription start site (TSS, page 13, line 1).

Addressed

2. *In figure legends please define error bars.*

Answer: We have now defined error bars standard deviation (SD) in individual figure legends, along with number of animals used in each figure panel.

Addressed

3. *Figure 4b unclear what the comparison is for each cohort.*

Answer: We have now modified the text of the **revised Figure 5b** to better explain the cohorts and comparison (**page 16, lines 10-16, and page 18, lines 3-10**). The cohorts include: a-humanized mice (circadian time points; control vs HCV), b-humanized mice (one time point, control vs HCV, and control vs DAA-cured), c-patients (uninfected/ no liver disease control vs HCV-infected), d-patients (uninfected/ no liver disease control vs HCV-cured; non-paired biopsy samples), e- patients (uninfected/ no liver disease control vs HCV-cured; paired biopsy samples), and f-patients (control vs NASH).

Authors have provided an explanation about each cohort, but in my opinion this clarification should be also included in heading of Figure 5 a and b (page 47, lines 4 and 8). In my opinion, it remains unclear what means "pool" in b-humanized mice, as authors indicate that "one time point" is included, but in Figure 5a and b "pool" is written and the name of the time point is not indicated.

4. *In figure 2e innate immunity pathways highlighted. How relevant is this in an immunocompromised model on an analysis of just hepatocytes?*

Answer: We completely agree with the reviewer. However, every cell is known to express part of the innate immune system as first line of defense to respond to stress and viral infection, and consistently this analysis (revised **Fig. 2d**) shows that the genes/pathways belonging to the hepatocytic innate immune response, i.e., cytokine signaling (TNF, IL6, Type 1 interferon) and their regulatory TFs (NFkB, STAT3) are activated following HCV infection.

Concerns have been clarified

5. *Extended figure 9 the names of the models (1-5) are not consistent with that used in the text and figure 2c.*

Answer: In the **revised Extended Figure 14** we have now corrected the names of models: maintained (unaltered; model 4), enhanced (gain; model 3), disrupted (loss; model 2), and shifted (altered; model 5).

Addressed

6. Could do with more clinical data for cohorts c and e in supplementary table 6

Answer: We have now provided additional clinical data for cohorts c and e in supplementary table 6.

Authors have provided additional clinical data for each cohort. Accordingly, there seems to be a slight error in Figure 5a and b, as the information in supp. Table 6 for cohort c (Taiwan-Holmes) include patients with fibrosis score F0-F2 but Figure 5 heatmaps showed that c cohort has F1-F4 fibrosis score. Additionally, some cohort are shown as pools and (c and d), while e is shown as individual patients, which is confusing.

7. Is HCV viral replication/ processes diurnal in this model?

Answer: We measured circadian levels of HCV titer from the serum of infected chimeric mice and found it to be higher during ZT16-ZT20 (active phase) (**Extended Data Fig. 12; page 11, lines 4-6**), which agrees with a previous study with limited number of patients showing higher HCV replication during active phase (Zhuang et al., Wellcome Open Research 2018, 3:96). However, further studies are necessary to confirm the significance of these observations.

Authors have resolved reviewer doubt

Reviewer #6:

Remarks to the Author:

This article by Mukherji and colleagues describes the utilisation of a mouse model grafted with human hepatocytes in an attempt to identify human hepatocytes rhythmic gene expression and how this expression is impacted by HCV infection. They also use a cell model to study a potential mechanism linking this alteration of rhythmic gene expression with a regulation of SIRT1 by HCV proteins. While the topic is of interest, many of the conclusions of the authors are not supported by experimental evidence.

A major issue consists in the utilisation of the humanised liver mouse model. In opposite to the claims of the authors, these mice have a strongly perturbed physiology. In addition to being immunocompromised, they have an altered rhythmic metabolism and physiology, as recently described by Delbes et al. (PMID: 37196091). It is really puzzling that authors cite this article to make the statement that “GH, STAT5 and PI3K/mTOR signaling pathways were also rhythmic in human hepatocytes” (pages 8 and 19) while this article shows that human hepatocytes do not respond to mouse GH and that the mTOR pathway is not rhythmic in humanised mice. Authors must correct this statement and acknowledge the limits of this model. In addition, they must compare their results with recent articles describing human liver gene expression using a different approach (PMID: 36730411; 36745672).

Another important issue is a global absence or improper use of statistics. Regarding rhythmic gene expression, there is no description of how authors determine differential rhythmic gene expression in human and mouse transcripts in humanised mice (fig. 1). Did they use DryR as in the HCV infection model? If yes, no detail is provided. If not, are authors comparing gene lists independently generated with JTK-cycle? If yes, according to the field (PMID: 29098954), this is not an acceptable method. In addition, authors did not explain if they differentiate human and mouse transcripts in the HCV infection model. Same problem for the ChIP-seq analysis: authors compare their human data with published mouse data. Why not using data from mouse hepatocytes in humanised mice? An additional problem is that when analysis was likely properly done, authors made wrong claims about the results. For example, authors claim that HCV infection perturbs the circadian clock. However, their own analysis shows that the rhythm of most clock genes is not impacted by the infection (all in model 4). Only BMAL1 shows an alteration in phase. Same for SIRT genes (SIRT1 and SIRT5 in model 4).

On the same idea, authors claim that HCV infection induces a “loss of rhythm” (page 11) of the human liver transcriptome while most of the rhythmic genes are not impacted (model 4, 78%) and almost the same number of genes lose (13%) or gain (8%) rhythm. According to this result, it is difficult to understand how the claimed perturbed rhythmic epigenome can lead to so little variation in gene expression. A potential explanation is that H3K9ac appears perfectly conserved in infected animals, in opposite to the claimed perturbed H3K27ac.

In addition, many claims made by the authors are not supported by any statistical analysis. For example, the change in SIRT1 protein levels (Fig. S19b,c) is not supported by any statistical analysis and appears unlikely based on the provided data. It is also not clear whether Actin is a proper control based on the high variability of the data. Same problem with PPAR γ (Fig 4g), MYC (Fig S15c), HCV titer (Fig S12), or all the Fig S18. Same for clock genes changes in Fig. 5a.

Finally, authors should refrain to refer to “circadian” expression while all the experiments were performed under a light/dark cycle.

POINT-BY-POINT RESPONSE TO THE REVIEWERS

Reviewer #4 (Remarks to the Author):

The authors have significantly improved the manuscript including in vitro validations of several claims and increasing the sample size to an acceptable number.

Answer: We thank the reviewer for the appreciation of the study and for the helpful comments.

Major Concerns:

1. It is unclear to me how the 2-replicates from the first experiment were chosen in the first half of the paper, why weren't all 3 of the original experiment included?

Response: Both referred animal experiments (**Series 1 and 2; Fig. 1 and Extended Data Fig. 2**), originally comprised livers of 3 human liver chimeric mice (HLCM) that were analyzed identically. To distinguish between transcripts originating from human and residual mouse liver tissue, we mapped sequence reads and annotated them to both the human and mouse genome (**Methods and Supplementary Table 2**). Post-annotation (mouse and human), we performed an unsupervised clustering to determine the phase of each circadian clock (CC) gene expression in the human hepatocytes of each HLCM (**Extended Data Fig. 2**).

We excluded one sample in Series 1, since only minimal reads were obtained for this animal (Extended Data Fig. 2; indicated in red). Nevertheless, the unsupervised/unbiased clustering of the remaining samples in Series 1 confirmed that 2 samples per time-point are comparable in phase of expression for all the 13 CC genes. We therefore selected these as the two closest samples for each circadian timepoint for the subsequent analysis. (Extended Data Fig. 2). Importantly, the same approach validated our findings in Series 2, where all the 3 samples/time-point are comparable for CC-genes expression. Taken together, we analyzed a total of 5 samples/time-point from two independent experiments, which enabled us to draw robust and arresting biological conclusions. In the revised manuscript, we now better explain the reasons why had to exclude one replicate from the analysis of Series 1 due to insufficient reads (page 7, lines 1-8).

Minor Concerns:

1. Extended Data Fig 4 -> the authors should perform a hypergeometric test (or Fisher's exact test) to determine the significance of this overlap. From my back of the envelop calculation it seems it is likely these are not significant.

Response: We thank the reviewer for this important point. Reviewer 6 requested to replace the JTK analyses in Fig. 1 by an analysis using the dryR algorithm as it is more suitable for intra-species comparisons (*Delbes et al., Sci Adv 2022*). To be consistent in the

manuscript, we have therefore subsequently reanalyzed all associated figures and extended datasets, including Extended Data Fig. 4.

As suggested by the reviewer, we now provided a hypergeometric test for our new dryR validated findings. We would like to highlight, however, that **dryR analyzes only orthologous genes** and Extended Data Fig. 4 compares rhythmic transcripts in human hepatocytes of HLCM with external data sets of mouse livers (*Koike et al., Science 2012*) and human livers (*Talamanca et al., Science 2023*). It should be furthermore noted that **both external data sets also include 'non protein coding' rhythmic genes, while our dataset includes solely rhythmic protein coding genes.** We clarified this in the revised manuscript (**page 8, lines 16-22, and page 41 lines 9-20**). These analyses also clarify and addresses the significance of the findings.

2. Extended Data Fig 5 -> the authors should double check their classification as "similar pathways" and "distinct pathways" because several "similar pathways" appear to be anti-correlated between human and mouse e.g. KE_VIBRIO_CHOLERAE_INFECTON.

Response: As suggested, we "double-checked" the classification of "similar pathways" and "distinct pathways". We classified them based on their relative 'peak' expression and allowed ± 1 circadian time point (± 1 ZT) to account for reaching the expression peak and its subsequent decrease during circadian oscillation. **To better illustrate this, the "KE_VIBRIO_CHOLERAE_INFECTON" pathway highlighted by the reviewer reaches its peak at ZT0 (human hepatocytes) and ZT4 (mice hepatocytes), which are separated by 1 time point, and hence classified as "similar pathway".** We have now clarified classification method including the circadian phase-specific expression (**page 9, lines 8-9**) providing now arresting conclusions.

3."Furthermore, we found that several of the zinc 9 finger TFs (ZNFs) were solely circadian in human hepatocytes (Extended Data Fig. 6)." - Wouldn't it be more accurate to say several of the ZNFs were only expressed in human hepatocytes as Ext. Fig 6 has grey squares representing no expression in the mouse data. How many of these ZNFs even have mouse orthologs? Are these simply human-specific genes?

Response: To address the reviewer's point on ZNFs/gene orthologues and the concerns of **Reviewer 6** regarding the use of JTK cycle (**previous Fig. 1**), **we have now consistently used dryR to compare the circadian transcriptome of human vs. mouse hepatocytes in the chimeric animals.** As emphasized previously, **dryR** is more suitable for **intra-species** comparison (*Delbes et al., Sci Adv 2022*), and **analyzes only orthologous genes present in both species** (human and mouse). Therefore, we **replaced JTK by dryR results in the revised Fig. 1 and Extended Data Fig. 6, and thus the revised Extended Data Fig. 6 includes ZNFs that are only expressed in both species.** We have modified the respective text to clearly address species-specificity (**page 7, line23, page 8 lines 1-16, and page 10 lines 10-13**).

4. Figure 2 - This whole figure is lacking information on statistical significance. Fig 2b should include a "" for which Log2FCs were significant and Figure 2D would benefit from a marker (e.g. a star) indicating which time points or which pathways are statistically significantly different and which are not.*

Response: We have now provided a detailed statistical analysis of results throughout the manuscript including Figure 2. We would like to mention that the analysis in **Fig. 2b** and **Extended Data Fig. 11b** have been **conducted on the same experiments and representing**

the circadian expression of different CC-genes either as heatmap (Fig. 2b) or as DESeq2 curves (Extended Data Fig. 11b). To maintain the visual clarity of Fig. 2b, we indicated statistical significance only in the Extended Data Fig. 11b. Note that, in the Extended Data Fig. 11b for *BMAL1* we found a significance of $FDR < 0.06$, which has been mentioned in the text (page 12, line 9).

We have now performed statistics for **Fig. 2d. Statistical significance ($P < 0.05$)** is represented as a **'bold' dot (increased diameters)** in individual figure panels and mentioned in figure legends.

Overall, for DESeq2 curves in the manuscript we have now represented statistical significance ($P < 0.05$; as bold dots) and specified this in the respective figure legends.

Reviewer #5 (Remarks to the Author): Please see attachment.

Below you will find in blue my opinion on the authors' response to reviewer 1.

Response: We thank the reviewer for the critical review of our manuscript and for the helpful comments. In the revised manuscript we have fully addressed all points by Reviewer 5 and highlighted our response within the point-by-point response as follows: To maintain clarity, we responded following the comments from the Reviewer 5 (questions indicated in blue). Our previous answers to the comments of Reviewers 1 and 2 are indicated in grey.

Major comments. 1. This is a descriptive study based on a single experiment with the authors building their conclusions on multiple complex bioinformatic analyses of transcriptomics data. For key pathways and disease processes highlighted I would like further validation. For example, a) confirmation at the protein level either using western blots or proteomics and confirmation of changes in downstream pathways. b) Key associated processes (steatosis and fibrosis) could also be confirmed histologically. c) For ChIP-seq data I would like to see validation using at least one additional marker of histone acetylation. d) For the authors proposed mechanism behind HCV induced alteration of circadian pathways I would like to see further proof experiments e.g. rescue by manipulation of SIRT1 or manipulation of the HCV virus to decipher mechanism. e) I would also like to see validation of the finding of incomplete reversal of HCV associated changes in CCcontrolled pathways. Could this be done in the human liver chimeric mouse model by treatment with anti virals?

Answer: As suggested, we addressed these points raised in the revised manuscript as follows:

a. To evaluate key pathways and disease processes altered following chronic HCV-infection of humanized liver chimeric mice (HLCM) at the protein level we validated oncogene MYC (driver of chronic liver disease), and SIRT1 (regulator of CC and metabolic pathways). Consistent with our transcriptomic analyses, the immunohistochemistry detected very low levels of MYC protein in the control liver, but its level was increased in the HCV-infected livers (Extended Data Fig. 15). Our analyses are consistent with studies noting minimal MYC expressed in control liver and its increased levels only in diseased state (Meyer and Penn, Nat Rev Cancer 2008, 976-990, and Buendia et al., Gastroenterology 2012, 214-218). Furthermore, we performed western blot

analysis which showed circadian expression of the SIRT1 in control liver and its reduction in HCV-infected livers (Extended Data Fig. 19b, c).

R5: In my opinion, the reviewer's concerns have been resolved. Authors have provided western blot to confirm changes in downstream pathways in HLCM.

Response: Thank you very much.

b. It is established that due to a lack of adaptive immune cells (T- and B-cells), in human liver chimeric mice, HCV infection leads only to mild fibrosis without significant steatosis (Mercer et al., Nat Med 2001, 927-933, Bissig et al., J Clin Invest 2010, 924-930, and Washburn et al., Gastroenterology 2011, 134-1344). Consistently, our histological analyses revealed mild fibrosis without any detectable steatosis (Extended Data Fig. 16a-c).

Authors have provided CK8-18 staining, but In HCV-infected HLCM liver, no steatosis was detected by histology, while modest but significant liver fibrosis were detected by Sirius red staining at different circadian time points and quantification of collagen positive areas (CPA).

Response: Thank you very much.

c. As suggested, we have now provided additional ChIP-seq data using a different marker of transcriptional activation: Histone 3 lysine 9 acetylation (H3K9ac), which consolidates our previous findings using H3K27ac on the HCV-induced epigenetic deregulation (Fig. 3c, d, Extended Data Fig. 17b and Supplementary Tables S3, S5). The new results are indicated (page 12, lines 21-23, page 13, lines: 1-2, 5-7 and 8- 15).

Authors included an additional marker of histone acetylation, the H3K9ac. They profiled how the HCV infection modifies both, the H3K9ac and H3K27ac in liver from infected and non-infected HLCM. However, in my opinion authors should also comment these new results in page 9, within the "Circadian regulation of transcription factors and enhancer regions impact pathways relevant for liver homeostasis". Here authors comment that initially they focused on H3K27ac, but they did not comment anything about the new data about H3K9ac, which in my opinion should be addressed in page 9.

Response: As suggested, we included a discussion on the H3K9ac ChIP-seq results in the revised manuscript (page 14, line 23, and page 15, lines 1-4).

d. We have now performed extensive mechanistic studies in well-established cell culture models of HCV replication and infection to address the role of HCV proteins in perturbed expression of the SIRT1 and key CC-gene BMAL1. These include CRISPR-Cas9 guided KO of SIRT1, pharmacological studies using SIRT1 agonist, and luciferase reporter assays. First, we first confirmed the key role of SIRT1 in the regulation of BAML1. Secondly, we identified the underlying molecular mechanism of HCV non-structural proteins suppressing SIRT1 expression by inducing PPAR γ (Fig. 4). These new results are now discussed (page 14, lines 19-23, page 15, lines 1-23, and page 16, lines 1-4).

Authors have provided new mechanistic experiments regarding the SIRT1 and key CC-gene BMAL1.

Response: Thank you very much.

e. To address the question of incomplete reversal of HCV associated changes in CC-controlled pathways in human liver chimeric mice treated with antivirals, we analyzed our previously published datasets (Hamdane et al., *Gastroenterology* 2019, 2313-2329). Our new analyses confirmed that the HCV-perturbed expression of CC-genes and key CC-controlled pathways driving chronic liver disease and HCC in the livers of humanized liver chimeric mice remain altered following viral cure (Fig. 5a, b; cohort b).

Authors have used a previously published results (ChIPmentation-based ChIP-Seq on liver tissue with H3K27ac antibody) related to 8 patients with chronic HCV infection cured by DAA treatment and apparently 5 treated with DAAs mice. In my opinion this information should be clarified as according to results showed in Fig. 5a, pooled data of HCV-infected and -cured humanized mice are showed, as well as pooled data of c, d and f patient cohorts. Have the authors provided any statistical analysis to show that there are no significant differences regarding HCV-perturbed expression of CC-genes following viral cure with DAAs?. These results should be shown. Additionally, information about what is Pat1, Pat 2 and Pat3 should be indicated in figure legend 5.

Response: As suggested, we have clarified these various points on different cohorts in the revised manuscript (page 18, lines 1-16) as follows:

We have now stated that analyses of CC genes for **cohorts b, c, and d** (Fig. 5a, b) **were performed from data of our previously published work** (*Hamdane et al., Gastroenterol* 2019) (page 18, lines 13-15).

We are now providing a **statistical analysis for the experiments using humanized mice (cohort b) in the new supplementary Table 7 (individual p values) confirming** that the observed transcriptional changes induced by HCV infection were not significantly different from cured samples. **Thus, changes in expression of CC-genes indeed persist after viral cure with DAAs (Page 19, lines 4-7).**

For the **different patient cohorts (c, d, e, and f; Fig. 5a)**, however, **a statistical analysis was not feasible as they followed different procedures for library preparation and RNA-seq. (as noted by the reviewer).** We have now **toned down our statement (Page 18, lines 18-20)**, and clarified the inability to perform direct statistical test as a **'limitation'** in the revised text (**Page 20, lines 11-13**). We have also provided **additional information about Pat 1-3** in legends (**Page 50, lines 6-7 and Page 50, lines 13-14**).

2. The robustness of the data from the human liver chimeric mouse experiment is questionable with only two biological repeats (n=2). I would like to see ideally n=5, but at the very least an n=3. The initial experiment had an n=3, can the authors do the analysis on this without selecting the two closest samples? I note the error bars on most the graphs are large. I would also like some statistical tests for significance in the differences highlighted .

Response : We have now performed an additional set of animal experiments with n=3 mice/group/ circadian time point. This new experiment has been used in the revised manuscript and key results from both the experiments are congruent (Fig. 1, 2 and Extended Data Fig. 1-19). Importantly, as suggested by the reviewer in this new experiment, we did not "select" samples

(Extended Data Fig. S2 and S9). With the addition of this new batch of animals we increased the robustness of our data, and the gene expression curves represent SD of 5 animals (5 chimeric mice/group/ circadian timepoint). For further pathway analyses we chose only those circadian pathways which showed FDR.

Here the authors have increased up to 3 the number of animals to increase robustness. However, they do not show any statistical results of DESeq2 to demonstrate differences commented by the reviewer.

Response: As suggested, we have now provided statistics for DESeq2 curves in the revised manuscript. These include genes and pathways comparing changes in control and HCV-infected human hepatocytes.

However, we would like to emphasize that a direct statistical analysis for the DESeq2 curves comparing transcript levels of two different species (human vs. mice) was avoided due to species-specific levels of gene expression. Importantly, as suggested by Reviewer 6 we have now used dryR to reanalyze Fig. 1 and Extended Figures 4-7. The dryR algorithm incorporates its own statistical test for different models of genes (Fig. 1c and Supplementary Table 3).

3. There are several problems with the patient biopsy cohorts. The controls for cohorts b and d are subjects without liver disease. The changes described may not be due to HCV but due to fibrosis/ chronic liver disease. Indeed there is evidence (in mice) that CCL4 induced fibrosis is associated with altered circadian rhythm of clock genes (doi: 10.1016/j.febslet.2010.03.019). The appropriate comparison would be patients with fibrosis/chronic liver disease from a non-viral etiology (e.g. NAFLD). I would also like more information on how RNA was obtained from the patient biopsy samples. Was RNA extracted from whole tissue or were hepatocytes isolated first? If the RNA was extracted from whole tissue then the additional cell types (immune cells, cholangiocytes etc) which are not in the human liver chimeric mouse analysis are a confounder when making the comparison.

Response : As suggested, we have now added transcriptomic data of NASH patients which exhibit alterations of CC genes and CC-pathways expression (Fig. 5a, b; cohort f). We would like to emphasize that a recent study (Jouffe et al.; PNAS 2022, Vol119, 10.e2200083119) noted perturbed CC-genes expression in NAFLD patients without fibrosis. Furthermore, our recent studies (J Hepatol-S-23-01054-R1, and J Hepatol 78, S77-78) also found CC-perturbation in NAFLD/NASH patients. Thus, CC-disruption seems to be an integral part of liver disease of different etiologies. We have now modified the text accordingly (page 18, lines 2-10). As the reviewer can appreciate that most of the samples are 'fine needle' biopsies and from infectious origin with barely enough materials making hepatocyte isolation technically challenging, we utilized whole RNA for these experiments. However, our chimeric mice studies do confirm an effect of HCV-infection on human hepatocytes.

Transcriptomic data of NASH patients have been included. However, they refer to heatmap figures to show differences between group of patients, but do not show statistical test (DESeq2 results) to confirm the visual trends observed in heatmaps. Additionally, according to authors "Our analyses indicate that compared to healthy controls both HCV 6 and NASH patients display perturbations of CC gene expression (Fig. 5a, cohorts c and f).(lines 5-6 page 18)", however this

figure do not show statistical differences of these comparisons for each gene displaying perturbations of expression. Regarding how the RNA was obtained from the patient biopsy samples, authors have resolved the reviewer concern about the RNA isolation procedure. However, they have not included this clarification in the text. This information should be clearly stated in material and method (page 34, line 4), as it has been clarified in results section (page 12 lines 1- 15) Additionally, it should be commented how the different library kit used for HLCM samples (TrueSeq stranded) and human liver samples (NEBNext) could introduced some bias in the in the results.

Response: As pointed out in the response above RNA-sequencing of patient cohorts (cohort c-f) were performed with different techniques and platforms from collaborating centers. This precludes a ‘direct’ statistical comparison between them. This limitation has now been discussed in the revised manuscript (Page 20, lines 11-13). We would like to highlight a recent study however, reporting a disruption of CC-genes expression in NASH patients (Jouffe et al.; PNAS 2022, Vol119, 10.e2200083119).

As suggested, we have now provided clarifications to the **materials and methods** section regarding the **limitations of sample collection from patients (Page 39, lines 8-10)** and discussed how different methods of library preparation may introduce a bias in the interpretation of results (Page 20, lines 11-13).

4. The manuscript needs more details of the human liver chimeric mouse model. Cryopreserved hepatocytes were obtained from BD biosciences, however if possible some basic clinical details of the source of these cells and their phenotype e.g. patient age, indication ? malignant? Also, information on the handling of these hepatocytes should be included. If they were defrosted and immediately implanted there are potential issues with stress and temperature shock, especially given cold temperatures have profound effects on clock genes (doi: 10.15252/emj.2020105604). If they were in culture for a prolonged period of time this too could be problematic as hepatocytes rapidly dedifferentiate in culture with hepatocyte-determining transcripts being down-regulated within hours (doi: 10.1046/j.1365-2613.1998.00083.x). I note there is a difference in the amount of repopulation of human hepatocytes in the control vs HCV groups (human albumin 14,203,835 vs 9,408,337 p=%HS 0.74 vs 0.53 p

Response : We have now provided more details on the chimeric mice and the PHH in the methods (page 32, lines 19-22, page 33, lines 1-2). We would like to emphasize that our engraftment procedure is state-of-the-art and identical to a recently published study on human hepatocytic clock in chimeric mice model (Delbés et al., Sci. Adv. 2023), thus suggesting this procedure does not lead to ‘potential issues with stress and temperature shock’ on clock genes.

Authors have provided more information on the human liver chimeric mouse model. Additionally, the referee was concerned about the difference in the amount of repopulation of human hepatocytes in the control vs HCV groups. Instead of commenting on these references, the authors have provided a set of additional experiments with a higher similarity of humanization level. Authors have confirmed that both, control and HCV-infected mice were engrafted with the same batch of PHH (page 33, line 1). Regarding the two independent RNA-seq analysis, authors comment that no significant differences in human-specific total reads were found between control and HCV-infected mice. However, any statistical test has been provided to prove this. In fact, I performed independent t-test analysis for each RNA-seq series 1 and 2, and significant statistical differences are found between both groups (0.001 and 0.046). To compare total number of reads

in the two experiments (n=18 series 1 and n=18 series 2) a previous normalization with all samples should be performed. Additionally, a significant different number of total reads were obtained for each series of RNA-seq, and therefore this different be accounted (different batch) to verify that no significant different were found between controls and HCV-infected mice.

Response: In the revised manuscript, we have now performed an additional set of re-analyses with control and HCV-infected mice, and in both experiments the degree of humanization in control was between 65-70% (Extended Data Fig. 10a). The degree of humanization in chimeric liver mice is known to be reflected by secretion of human albumin (Mercer et al., *Nat Med* 2001; Bissig et al., *J Clin Invest* 2010; Grompe and Strom, *Gastroenterology* 2013; Maily et al., *Nat Biotechnol* 2015; Delbes et al., *Sci Adv* 2022). At the timepoint of infection, we found similar (if not slightly higher) human albumin levels in mouse plasma samples of the group of HCV-infected mice; however, those levels dropped during HCV infection (Extended Data Fig. 10c) explaining the significant lower repopulation pointed out by Reviewer 5 at the endpoint. The loss of hepatocytes during chronic HCV infection in the human liver chimeric mice has been described previously (Maily et al., *Nature Biotech* 2015). It is well described that chronic HCV infection creates cellular stress predisposing to chronic liver disease (Lupberger et al., *Gastroenterology* 2019), which are confirmed by our histological (CK18 staining; Extended Data Fig. 10b) and transcriptomic analyses (Fig. 2d). Importantly, the transcriptomic analyses of control and HCV-infected mice involved sequenced reads originating specifically from human hepatocytes. This was achieved through our mapping strategy which avoids artifacts from different humanization levels in control and HCV-infected samples. For clarification, we have better emphasized these points in the revised manuscript (page 11, lines 17-22, and page 12, lines 1-4).

5. The authors have presented the majority of the data in the main figures as heatmaps. Whilst these are visually pleasing I find them very difficult to interpret separate to the main text. For some even after reading the main text and repeatedly studying the figures I was unable to see the same patterns/ interpretation. I am concerned that general readers will struggle. Examples a). I could not interpret figure 4c, not understanding why each variable appears to be repeated and did not understand the meaning of the brown and blue bars. b). In figure 2c the meaning of the target plots and different colour scheme (green alone, red alone, brown and red and green) is not explained. c). The different colored heatmaps for mouse vs human and HCV vs control make it difficult for the reader to make direct comparisons themselves. I would advise using the same colours as they are clearly labelled. I would also advise the authors displaying all the genes in 1b and 2b in extended figures 2 and 7 respectively as trends more easily seen.

Response : As, suggested by the reviewer we have now revised the color schemes of the heatmaps for more clarity.

a. We have now explained the interpretation of previous Fig. 4c in a more lucid manner in the main text and the figure legends. "Importantly, we observed changes in gene set enrichment index (GSEI) which indicates 'magnitude of correlation' with overall survival of the patients according to HCVclock classification (Fig. 5d, e), suggesting its possible clinical relevance" (page 19, lines 1-3). Figure legend 5c now reads: "Correlation between induction of the CC-oscillator signatures and clinical demographics and our previously reported Prognostic Liver Signature (PLS) in 216 early-stage (i.e., Child-Pugh class A) HCV cirrhosis patients. The signatures were defined as

genes with > 4-fold over-expression in either HCV-infected or control hepatocytes in early or late time points in each of the 4 representative models (i.e., loss, gain, altered, and unaltered). All pairwise correlations between the variables (correlation matrix) as shown in rows and columns. The correlation matrix was clustered to depict groups of variables sharing similar patterns of correlation.” The brown, deep blue and light blue color bars indicate the presence of the 3 major correlation clusters. Figure legend 5d now reads: “Classification of the early-stage cirrhosis patients by the CC-oscillator signatures associated with clinical featured related to liver disease severity (Supplementary table 7), Magnitude of gene signature/set modulation is shown in each patient (in each column) in the cohort. The induction/suppression of the circadian clock gene sets (shown in bottom half of the panel) was determined by Gene Set Enrichment Index (GSEI) as previously described (Nakagawa et al., Cancer Cell 2016, 879-890). The PLS and HCVclock risk predictions were determined by using Nearest Template Prediction (NTP) algorithm (Hoshida, Plos One 2010, e15543). For the PLS, normalized expression levels or the PLS member genes were used for the NTP analysis as previously described (Hoshida et al., N Engl J Med 2008, 1995-2004, Hoshida et al., Gastroenterology 2013, 1024-1030). For the HCVclock risk prediction, we used the GSEIs of the circadian clock gene sets for the NTP analysis.”

Authors have clarified the figure 5d and e (previously figure 4d).

Response: Thank you very much.

b) We have revised the text explaining the meaning of the target plots and different color schemes for Fig. 2c (Page 11, lines 11-15).

Figure 2c has been clarified.

Response: Thank you very much.

c) We have now used same-colored schemes for the heat maps mouse vs human (Fig. 1b and 1d). All the genes shown previously in Figs. 1b, 2b are now also present in the revised Extended Figures 3a and 11b.

Reviewer wanted to see same coloured schemes for either mouse vs human comparison and HCV vs control, which has not been addressed: Figure 1b: blue for human green for mouse. Figure 1c: blue for series 1 and green for series 2 of human hepatocytes in HLCM. Figure 1 e: which according to authors shows “Circadian expression pattern of transcription factors (TFs) in human hepatocytes (red) and in 11 residual murine (green) cells in HLCM (n=5 HLCM/ circadian timepoint), as predicted by JTK.” However, the image showed 7 collapsed heatmaps for each ZT for series 1 and 2. Figure 1d: black for human and mouse, and blue to red for pathways. Authors should use the same colour palette for heatmaps of genes and pathways. Dark blue to yellow are used for genes in figure 1, but blue to red in figure 5. On the other hand, figure 5a and 5 b, used same colour for genes and pathways.

Response: We have now simplified and depicted heatmaps of identical colors for genes (Fig. 1b and 5a) and pathways (Fig. 1d and 5b) including the use of the same color scheme for human vs. mouse as suggested by the reviewer.

Only descriptive data are showed in Figs. 1b, 2b and Extended Figures 3a and 11b, no statistical differences for each gene is showed.

Response: We are now providing a thorough statistical analysis throughout the revised manuscript with few exceptions where a statistical analysis was not possible: Fig. 1b and Extended Figure 3a compare whether the ‘phase of expression of core CC-genes’ are identical (not the expression levels) in transplanted human hepatocytes and residual murine hepatocytes.

As Fig. 1b and Extended Figures 3a are cross-species analysis, where individual genes are expressed in different levels (Delbes et al., Sci Adv 2022), hence a direct statistical comparison of expression levels is not advisable.

Moreover, to maintain clarity of the figure we provided the statistics for Fig. 2b (heatmap of log2FC) in the corresponding DESeq2 curves in Extended Figure 11b (DESeq2 curves), which show essentially the same information (expression of CC-genes in the human hepatocytes) in control and HCV-infected mice.

6. The authors do not explain how (or why) HCV brings about the observed circadian changes or the associated alterations in H3K27 acetylation. Without mechanistic investigation and then interventional work the study does not progress sufficiently beyond simple observation. As examples; is this an effect of a particular viral protein or transcript, is there increased expression of a H3K27 acetyl transferase or a demethylase and can this type of effect be associated with the activities of a particular viral protein? Answer: Our results indicate that the HCV-induced overall increase in H3K27ac and H3K9ac (Fig. 3) originates from a multilayered approach. An overall increase in histone acetylation likely is the result of either an increase in histone acetylase activity/levels or decreased histone deacetylation activity/levels. Of the known histone acetylases, we found HCV-infection increase the transcript levels of only CREBBP (CBP) (Extended Data Fig. 18). We found that HCV does not significantly affect the level of canonical histone deacetylases (Extended Data Fig. 18). However, HCV affects the transcript levels of multiple histone deacetylases belonging to the Sirtuin family, including SIRT1 (Fig. 3h). Importantly, we found that in control liver the SIRT1 protein display circadian rhythmicity with a peak in the rest phase, and HCV-infection leads to substantial reduction in peak SIRT1 expression (Extended Data Fig. 19 b, c). In the revised manuscript we revealed that HCV-infected cells and chimeric mice liver promote the expression PPAR γ , which is a transcriptional repressor of SIRT1 (Han et al., Nucleic Acid Res 2010, 7458-7471, and Yu et al., J Biol Chem 2005, 13600- 13605), which correlates with a reduced SIRT1 expression (Fig. 4). We demonstrate that HCV proteins NS4 and NS5A promote PPAR γ levels while decreasing levels of the epigenetic modifier SIRT1 (Fig. 4). In perturbation studies, we established a functional link between SIRT1 and BMAL1 expression. Taken together, we demonstrate that HCV employs an elaborate and multistep strategy to alter the acetylation level of histones on a global scale, in which reduction in the levels and/or activity of the deacetylases (Sirtuins) play a major role. However, we believe that further undiscovered mechanisms (e.g., activity of CREBBP, and deregulated expression of circadian TFs) could also be contributing towards altering the epigenome, which remains beyond the scope of this present study and will be addressed in future investigations.

The new results are shown in new Fig. 3, 4 and Extended Data Fig. 18, 19 and discussed in the revised manuscript (page 14, lines 3-23, page 15, lines 1-23, page 16, lines 1-4, page 21, lines 4-22).

Authors show different observations on transcript levels regarding HCV infection to explain how HCV brings about the commented alterations in H3K27 acetylation. However, first of all these data are only showed as normalized counts and no statistical results from DESeq2 are shown (only bars representing SD). Most of the bars overlap in Extended Data Fig. 18, so therefore no significant different should be present. However, Fig. 3h shows also overlapping in all the ZT for SIRT1, but according to authors HCV affects the transcripts levels of SIRT1, this seems contradictory. If authors provide statistical test results should be easier to understand. Nevertheless, authors provide an additional set of experiments to demonstrate that HCV proteins decrease levels of SIRT1 (Figure 4).

Response: In our study we **identified circadian variations in H3K27ac levels** in liver cells of human and WT mice, which are **abrogated in its oscillation during HCV infection**. Aiming to unravel the molecular mechanism, **we show that HCV infection suppresses expression of SIRT1 protein**, which is a known **regulator of histone acetylation (Fig. 4 and Extended Data Fig. 18b-c)**. Indeed, **transcript levels of most epigenetic regulators in HLCM “most of the bars do overlap” (revised Extended Data Fig. 17; previously Extended Data Fig. 18)**, which we did discuss as “In HLCM, no significant transcriptional change of key histone acetylases (*EP300*, and *CREBBP*) or deacetylases of the *HDAC* or *SIRTUIN* families were observed upon HCV-infection” (**page 15, lines 18-20**).

As suggested by the reviewer, we have now performed the statistical analyses of DESeq2 curves in Fig. 3h and revised Extended Data Figure 17 (previously Extended Data Fig. 18), confirming that the **observed transcriptional trend of reduced SIRT1** (and for other epigenetic regulators) **in HCV-infected livers were not significant**. We have now clearly mentioned this in the revised text (**page 15, lines 18-20**).

We believe that **along with the reduced SIRT1 protein levels** further undiscovered transcriptional and post-transcriptional mechanisms (**e.g., deregulated TFs and epigenetic regulators expression and activity**) **could lead to the observed phenotype**, which are beyond of the scope of this study and will be addressed in future investigations. **We have discussed these points in the revised manuscript (page 23, lines 19-23 and page 24, lines 1-3)**.

Minor comments.

1. Some abbreviations such as PPH and TSS have not been defined and will not be understood by a general reader.

Response : We have now explained the abbreviations primary human hepatocytes (PPH; page 6, line 6), and transcription start site (TSS, page 13, line 1).

Addressed

Response: Thank you very much.

2. In figure legends please define error bars.

Answer: We have now defined error bars standard deviation (SD) in individual figure legends, along with number of animals used in each figure panel.

Addressed

Response: Thank you very much.

3. Figure 4b unclear what the comparison is for each cohort.

Answer: We have now modified the text of the revised Figure 5b to better explain the cohorts and comparison (page 16, lines 10-16, and page 18, lines 3-10). The cohorts include: a-humanized mice (circadian time points; control vs HCV), b-humanized mice (one time point, control vs HCV, and control vs DAA-cured), c-patients (uninfected/ no liver disease control vs HCV-infected), d-patients (uninfected/ no liver disease control vs HCV-cured; non-paired biopsy samples), e-patients (uninfected/ no liver disease control vs HCV-cured; paired biopsy samples), and f-patients (control vs NASH).

Authors have provided an explanation about each cohort, but in my opinion this clarification should be also included in heading of Figure 5 a and b (page 47, lines 4 and 8). In my opinion, it remains unclear what means “pool” in b-humanized mice, as authors indicate that “one time point” is included, but in Figure 5a and b “pool” is written and the name of the time point is not indicated.

Response: “Pool” for cohort b refers to the combined transcriptomic results from the humanized mice in either control vs HCV-infected or control vs HCV-cured groups (Hamdane et al., Gastroenterology 2019). The mice in cohort b were all sacrificed around same circadian time-point in rest phase. To better explain, we now provide more details about the cohorts and the used terminology in the **respective figure legends (page 50, lines 2-7 and page 50, lines 10-14; previously page 47, lines 4 and 8).**

4. In figure 2e innate immunity pathways highlighted. How relevant is this in an immunocompromised model on an analysis of just hepatocytes?

Answer: We completely agree with the reviewer. However, every cell is known to express part of the innate immune system as first line of defense to respond to stress and viral infection, and consistently this analysis (revised Fig. 2d) shows that the genes/pathways belonging to the hepatocytic innate immune response, i.e., cytokine signaling (TNF, IL6, Type 1 interferon) and their regulatory TFs (NFκB, STAT3) are activated following HCV infection.

Concerns have been clarified.

Response: Thank you very much.

5. Extended figure 9 the names of the models (1-5) are not consistent with that used in the text and figure 2c.

Answer: In the revised Extended Figure 14 we have now corrected the names of models: maintained (unaltered; model 4), enhanced (gain; model 3), disrupted (loss; model 2), and shifted (altered; model 5).

Addressed

Response: Thank you very much.

6. Could do with more clinical data for cohorts c and e in supplementary table 6

Answer: We have now provided additional clinical data for cohorts c and e in supplementary table 6.

Authors have provided additional clinical data for each cohort. Accordingly, there seems to be a slight error in Figure 5a and b, as the information in supp. Table 6 for cohort c (Taiwan-Holmes) include patients with fibrosis score F0-F2 but Figure 5 heatmaps showed that c cohort has F1-F4 fibrosis score. Additionally, some cohort are shown as pools and (c and d), while e is shown as individual patients, which is confusing.

Response: We would like to clarify that in Fig. 5a and 5b, cohorts c and d were used from our previously published work (Hamdane et al., Gastroenterology 2019) which is clearly stated in the revised manuscript on page 18, lines 13-15.

The Taiwan HCV cohort (Table 6) is shown as individual patients in “cohort g” (Extended Data Fig. 19b). The patients in Taiwan (Table 6, cohort g of Extended Data Fig. 19b) were at F0-F2 prior to their treatment and were at F1-4 post-viral cure. The RNA-seq from cohort g was performed from viral-cured samples. As, cohort e (Japanese) and cohort g (Taiwan) were provided by our collaborators, while cohorts b, c, d, and f used datasets of our previously published work (Hamdane et al., Gastroenterology 2019, Juehling et al., Gut 2021), we chose to represent external cohorts as individual patients for highlighting individual variations. As suggested by the reviewer we have now better explained the cohorts in the revised manuscript (page 50, lines 2-7, and page 50, lines 10-14).

7. Is HCV viral replication/ processes diurnal in this model?

Answer: We measured circadian levels of HCV titer from the serum of infected chimeric mice and found it to be higher during ZT16-ZT20 (active phase) (Extended Data Fig. 12; page 11, lines 4-6), which agrees with a previous study with limited number of patients showing higher HCV replication during active phase (Zhuang et al., Wellcome Open Research 2018, 3:96). However, further studies are necessary to confirm the significance of these observations.

Authors have resolved reviewer doubt.

Response: Thank you very much.

Reviewer #6 (Remarks to the Author):

This article by Mukherji and colleagues describes the utilisation of a mouse model grafted with human hepatocytes in an attempt to identify human hepatocytes rhythmic gene expression and how this expression is impact by HCV infection. They also use a cell model to study a potential mechanism linking this alteration of rhythmic gene expression with a regulation of SIRT1 by HCV proteins. While the topic is of interest, many of the conclusions of the authors are not supported by experimental evidence.

1. A major issue consists in the utilisation of the humanised liver mouse model. In opposite to the claims of the authors, these mice have a strongly perturbed physiology. In addition to being immunocompromised, they have an altered rhythmic metabolism and physiology, as recently described by Delbes et al. (PMID: 37196091). It is really puzzling that authors cite this article to make the statement that “GH, STAT5 and PI3K/mTOR signaling pathways were also rhythmic in

human hepatocytes” (pages 8 and 19) while this article show that human hepatocytes do not respond to mouse GH and that the mTOR pathway is not rhythmic in humanised mice. Authors must correct this statement and acknowledge the limits of this model.

Response: We used human liver chimeric mice (HLCM) as it is widely accepted to study aspects of human chronic liver disease caused by viruses and metabolic stress (Mercer *et al.*, *Nat Med* 2001; Bissig *et al.*, *J Clin Invest* 2010; Grompe and Strom, *Gastroenterology* 2013; Maily *et al.*, *Nat Biotechnol* 2015; Hamdane *et al.*, *Gastroenterol* 2019; Lupberger *et al.*, *Gastroenterology* 2019; Rohlen *et al.*, *Sci Transl Med* 2022). We agree that every model has specific limitations as pointed out by the reviewer. We have therefore better discussed these limitations in context to the referred publication from Delbes and coworkers in the revised manuscript (page 22, lines 10-13).

As suggested, we have now corrected our previous statement “GH, STAT5 and PI3K/mTOR signaling pathways were also rhythmic in human hepatocytes” to “Consistent with a recent study²⁷ we also observed that key genes regulating growth hormone (GH), STAT5 and mTOR signaling pathways were non-rhythmic (model 1) in transplanted human hepatocytes (Supplementary Table 3)” (page 9, lines 13-15).

2. In addition, they must compare their results with recent article describing human liver gene expression using a different approach (PMID: 36730411; 36745672).

Response: As suggested by the reviewer, we have now compared the cycling genes identified in human hepatocytes of HLCM with those of Talamanca and co-workers (PMID 36730411) and listed the matching genes (Supplementary Table 3), and also provided a Venn diagram of the overlap (Extended Data Fig. 4b). However, a comparison with the data from Wucher and co-workers (Wucher *et al.*, *PLoS Biol* 2023, PMID 36745672) was not possible as this paper mostly addressed gene expression patterns in brain and gonadal tissues but does not provide information for liver diurnal transcriptome.

3. Another important issue is a global absence or improper use of statistic.

Response: We have revised the used statistical tests and are now providing a thorough statistical analysis throughout the revised manuscript and better indicated this in respective figures and legends.

4. Regarding rhythmic gene expression, there is no description of how author determine differential rhythmic gene expression in human and mouse transcripts in humanised mice (fig. 1). Did they used DryR as in the HCV infection model? If yes, no detail is provided. If not, are authors comparing gene list independently generated with JTK-cycle? If yes, according to the field (PMID: 29098954), this is not an acceptable method.

Response: As suggested by the reviewer, we have replaced the JTK analyses in Fig. 1 by an analysis using the dryR algorithm as it is more suitable for intra-species comparisons (Delbes *et al.*, *Sci Adv* 2022). To be consistent in the manuscript, we have therefore subsequently reanalyzed all associated figures and extended datasets, including Fig. 1c-f

and Extended Data Fig. 4-7 and Supplementary Table 3. As suggested, we have added details of dryR analysis in methods (page 41, lines 9-20).

5. In addition, authors did not explain if they differentiate human and mouse transcripts in the HCV infection model.

Response: HCV is very species specific and infects exclusively hepatocytes in humans and not murine cells (Mercer et al., Nat Med 2001; Bissig et al., J Clin Invest 2010; Grompe and Strom, Gastroenterology 2013; Mailly et al., Nat Biotechnol 2015; Hamdane et al., Gastroenterol 2019; Lupberger et al., Gastroenterology 2019). **Hence, in the infection studies we only analyzed gene expression in human hepatocytes.** This has been clarified (page 11, lines 12-13, and page 12, lines 5-7).

6. Same problem for the ChIP-seq analysis: authors compare their human data with published mouse data. Why not using data from mouse hepatocytes in humanised mice?

Response: The human liver chimeric mouse model is based upon repopulation of murine hepatocytes by engrafted human hepatocytes. The degree of humanization (repopulation rate) was 65-70%. This limited the required input for ChIP on the remaining murine cells. Moreover, available H3K27ac- and H3K9ac-specific antibodies for ChIP are not strictly species-specific which would have further confounded the analyses in the context of a largely overrepresented human part. Because of these reasons we focused on the analysis of published mouse datasets.

7. An additional problem is that when analysis where likely properly done, authors made wrong claims about the results. For example, authors claim that HCV infection perturb the circadian clock. However, their own analysis show that the rhythm of most clock genes is not impacted by the infection (all in model 4). Only BMAL1 show an alteration in phase. Same for SIRT genes (SIRT1 and SIRT5 in model 4). On the same idea, authors claim that HCV infection induces a “lose of rhythm” (page 11) of the human liver transcriptome while most of the rhythmic genes are not impacted (model 4, 78%) and almost the same number of genes lose (13%) or gain (8%) rhythm. According to this result, it is difficult to understand how the claimed perturbed rhythmic epigenome can lead to so little variation in gene expression. A potential explanation is that H3K9ac appears perfectly conserved in infected animals, in opposite to the claimed perturbed H3K27ac.

Response: We agree with the reviewer. However, we would like to mention that **2 transcription factors** (TFs) of CC-circuit **BMAL1** and **NFIL3** belong to **model 5** (altered; **Supplementary Table 4**), and the peak expression of **BMAL1-target DBP** (another TF) is reduced (**Extended Data Fig. 11b**). **We do agree with the reviewer that more genes belong to model 4 than in models 2, and 3.** We believe that models 2, 3 and 4 (**accounting for ~1000 genes**) including several TFs and enzymes which in combination **significantly influence** (directly and indirectly) **the alteration of disease relevant pathways (Fig. 2d)**. We have now toned down the words ‘**loss of rhythm**’ (previously Page 11) to ‘**deregulation**’ (page 13, line 9), and have revised the entire sentence (page 13, lines 8-10).

As suggested by the reviewer, indeed it is a distinct possibility that largely unchanged H3K9ac (affected during ZT8-ZT20), which happens in both gene body and promoter-enhancer regions nullifies some of the epigenomic changes e.g. for genes in model 4. Indeed, we have previously demonstrated that most significant transcriptomic changes associated with HCV infection in patients were linked to H3K27ac (Hamdane et al., Gastroenterology 2019). We believe that multiple molecular mechanisms underlie HCV-induced diurnal transcriptomic and epigenomic changes and need to be explored in future studies. Addressing the reviewers concern, we have now down toned some of the conclusions and discussed these possibilities (page 23, lines 19-23, and page 24, lines 1-3).

8. In addition, many claims made are not supported by any statistical analysis. For example, the change in SIRT1 protein levels (Fig. S19b,c) is not supported by any statistical analysis and appears unlikely based on the provided data. It is also not clear whether Actin is a proper control based on the high variability of the data. Same problem with PPARg (Fig 4g), MYC (Fig S15c), HCV titer (Fig S12), or all the Fig S18. Same for clock genes changes in Fig. 5a.

Response: As mentioned before, we have carefully revised the used statistical tests and are now providing a thorough statistical analysis throughout the revised manuscript and better indicated this in respective figures and legends. SIRT1 transcripts in human hepatocytes of chimeric mice liver showed a rhythmic expression (Fig. 3g, h). This led us to choose two representative timepoints (ZT0-early rest phase and ZT12-early active phase) in Extended Data Fig. 18b and 18c (previously Fig. S19b and S19c) and performed western blot analyses using all samples (3 chimeric mice/timepoint/group) from both Series 1 and Series 2. We have now performed WB analyses for human SIRT1 and β -tubulin in the humanized livers of chimeric mice from two independent infection experiments (Series 1 and 2) (Fig. S18b). Additionally, we have analyzed human CK18 (Fig. S18b), which is only expressed in human hepatocytes, reflecting repopulation of the analyzed liver section (Fig. S10b). Of note, as HCV only infects human hepatocytes the repopulation in chimeric livers is decreasing during chronic infection (see response to reviewer #5, point 4, pages 7-8). Importantly, since both anti-tubulin (GenTex; #GTX101279) and anti-actin (Chemicon; #MAB1501) antibodies are not specific for the human orthologs and detect both mouse and human liver proteins, this approach is not suitable to quantify the human SIRT1 expression relative to human liver protein present in a sample. To specifically quantify SIRT1 relative to the amount of human hepatocytes in the liver samples loaded on the gel we quantified human SIRT1 expression relative to human CK18 as a reference protein. This approach validated our observation of an abrogated SIRT1 oscillation in human hepatocytes of HCV-infected chimeric livers. Additionally, we have also provided stain-free gels (loading control for total proteins; Extended Data Fig. 18 and 25), which were used to detect human SIRT1, human CK18 and β -tubulin in chimeric mouse livers (page 56, lines 22-23, and page 57, lines 1-2). It should be noted that key findings related to the mechanism and SIRT1 were validated in additional infection models confirming that HCV-infection strongly reduces SIRT1 protein level (Fig. 4).

Fig. 4g: Indeed, PPARG transcript levels are not significantly different between control and HCV-infected samples (page 17, lines 13-14). However, validation experiments using Western blotting and immunofluorescence confirmed that HCV-infection increases PPARG protein levels (4h-i and Extended Data Fig. 23). We describe these additional validations in the revised manuscript on page 17, lines 13-17.

MYC (previously **Fig. S15c**): We have now **provided statistical test** comparing **MYC expression in control and HCV-infected HLCM liver** (revised **Extended Data Fig. 14c**; previously **Extended Data Fig. S15c**). We would like to highlight that our results are consistent with studies noting HCV-induced MYC expression (*Juehling et al., Gut 2021*).

Fig. S12: We performed this analysis to address Reviewer 5 (**minor comment 7**). As this, however, is **not the aim and scope of the manuscript**, we would like to emphasize that this result needs further investigation to explore this possibility (**page 12, lines 21-22**).

Fig. S18: Revised figures S17 (previously **Fig. S18**) we performed statistical test and modified the text accordingly (**page 15, lines 18-20**).

Fig. 5a: This figure analyzed CC-genes expression from patients and humanized mice liver. As the reviewer will understand that **paired biopsies of HCV-cured patient's samples are a rarity**. To circumvent this problem, we included samples from **different collaborating partners, who followed different techniques of library and sequencing platforms. This precludes a 'direct' statistical comparison between them**. We have now mentioned this as a limitation of our study (**page 20, lines 11-13**).

However, samples from **cohort 'b' were prepared and sequenced identically and we have performed the statistical test** (**Supplementary Table S7**).

9. Finally, authors should refrain to refer to "circadian" expression while all the experiments were performed under a light/dark cycle.

Response: Agreeing with the reviewer **in the revised manuscript at majority of places we have replaced 'circadian' with alternative words e.g., 'rhythmic', 'cycling' or 'diurnal' depending on the context**.

Reviewers' Comments:

Reviewer #4:

Remarks to the Author:

The authors have addressed all of my concerns, and I believe the manuscript is now acceptable for publication.

Reviewer #5:

Remarks to the Author:

Dear Editor, the authors have satisfactorily resolved each of the questions posed to them. I recommend their acceptance and publication

Reviewer #6:

Remarks to the Author:

While this revised article shows some improvement compared to the previous version, it falls short to solve many of the previous issues. In particular, the part related to the disruption (or not) of the circadian clock after HCV infection is very confusing or even misleading. Below are the main comments to the answers/manuscript.

- Circadian disruption:

Page 15: "We agree with the reviewer. However, we would like to mention that 2 transcription factors (TFs) of CC-circuit BMAL1 and NFIL3 belong to model 5 (altered; Supplementary Table 4), and the peak expression of BMAL1-target DBP (another TF) is reduced (Extended Data Fig. 11b)."

It is very bizarre to admit that there is very limited alteration of the circadian clock (only 2 genes with slightly altered phase) but include a figure (2b and Extended 11a) with the only purpose to show non-existing differences! This is likely because this "disruption" justifies figure 5a, but this is really misleading. This must be removed from the abstract and clearly stated in the text, as well as the fact that most rhythmic genes (80%) present no difference in rhythmicity.

Page 9: "We have now simplified and depicted heatmaps of identical colors for genes (Fig. 1b and 5a) and pathways (Fig. 1d and 5b) including the use of the same color scheme for human vs. mouse as suggested by the reviewer."

Figures 2B, extended 2B, 6, 11 are still using different colours.

- Human cohort data:

Page 5: "For the different patient cohorts (c, d, e, and f; Fig. 5a), however, a statistical analysis was

not feasible as they followed different procedures for library preparation and RNA-seq.”

Page 7: “As pointed out in the response above RNA-sequencing of patient cohorts (cohort c-f) were performed with different techniques and platforms from collaborating centers. This precludes a ‘direct’ statistical comparison between them.”

If the data obtained with this different cohorts are not comparable, they should not be compared! Considering the previous comment, this part must be removed from the manuscript if now additional validation of the comparison is provided.

- SIRT1 and PPARG expression:

Page 16: “This approach validated our observation of an abrogated SIRT1 oscillation in human hepatocytes of HCV-infected chimeric livers.”

Authors provide new results showing that SIRT1 expression is lower at ZT12 compared to ZT0 in control animals while no time difference is observed in infected animals. With only 2 time points, it is misleading to discuss “rhythmicity”. In addition, no statistical analysis is provided for the comparison between control and infected animals. According to the observed variability, there is likely no difference and, if any, this could not explain the difference in acetylation. Altogether, this “SIRT1 story” is still not supported by the data.

Page 16: “Fig. 4g: Indeed, PPARG transcript levels are not significantly different between control and HCV-infected samples (page 17, lines 10-11). However, validation experiments using Western blotting and immunofluorescence confirmed that HCV-infection increases PPARG protein levels (4h-i and Extended Data Fig. 23). We describe these additional validations in the revised manuscript on page 17, lines 11-14.”

This is a cell experiment, completely unrelated to the question. PPARG must be measured in the animal models to justify this conclusion. This also must be the case for BMAL1 protein.

- Additional comments to the manuscript:

- Authors now admit that the GH, STAT5, and mTOR pathways lost rhythmicity, referring to Supplementary Table 3. There is absolutely no analysis of these pathways in this table. What leads to this conclusion?
- Figure 2d and associated text: This figure is supposed to show the “disruption” of rhythmic pathways, but most pathways are not rhythmic and show only in difference in the level of enrichment. This must be more clearly described.
- Statistics for the difference of rhythmicity are still missing for histone modifications.
- Information about the human hepatocyte donors is missing.
- Repetitions of the descriptions of the models (pages 8 and 13).
- Authors still refer to “circadian” rhythmicity throughout the manuscript.

POINT-BY-POINT RESPONSE TO THE REVIEWERS

Reviewer 6: Summary

While this revised article shows some improvement compared to the previous version, it falls short to solve many of the previous issues. In particular, the part related to the disruption (or not) of the circadian clock after HCV infection is very confusing or even misleading.

Response: We thank the reviewer for the critical and constructive review of our findings and the helpful comments which allowed us to improve the manuscript. **We now resolved all points raised and modified the manuscript and figures accordingly.** To track changes, we provide a detailed point-by-point response below and marked changes in blue in the revised manuscript as stated as well in the point-by-point response.

(main comments to the answers/manuscript:)

- Circadian disruption:

Page 15: “We agree with the reviewer. However, we would like to mention that 2 transcription factors (TFs) of CC-circuit BMAL1 and NFIL3 belong to model 5 (altered; Supplementary Table 4), and the peak expression of BMAL1-target DBP (another TF) is reduced (Extended Data Fig. 11b).” It is very bizarre to admit that there is very limited alteration of the circadian clock (only 2 genes with slightly altered phase) but include a figure (2b and Extended 11a) with the only purpose to show non-existing differences! This is likely because this “disruption” justifies figure 5a, but this is really misleading. This must be removed from the abstract and clearly stated in the text, as well as the fact that most rhythmic genes (80%) present no difference in rhythmicity.

Response: As requested, we down toned the conclusions drawn from Fig. 2b and Extended Fig. 11 and **removed unaltered genes from Fig. 11b.** We removed the statement referring to clock from the abstract: “We show that hepatitis C virus (HCV) infection, a major cause of liver disease and cancer, *perturbs the transcriptome by altering the rhythmicity of more than 1500 genes*, and affects epigenome, leading to an activation of critical pathways mediating metabolic alterations, fibrosis, and cancer” (abstract). Furthermore, we also down toned the results section with the following statement: “*Even though HCV-infection perturbed the*

expression of only some CC genes (Fig. 2b), this perturbation was associated with a dysregulation of the transcriptomic oscillation of 22% of rhythmic genes (loss, gain, and altered rhythmicity) (Fig. 2 and Extended Data Fig. 13)” (page 21, lines 1-3).

As requested, we have **removed** the following statement: “*However, other key CC genes (CLOCK, PER1 and REV-ERB α /NR1D1) displayed **only overall tendencies of dysregulation** in infected hepatocytes (Fig. 2b, Extended Data Fig. 11a)*”. (Previously in page 12, lines 12-13).

Page 9: “We have now simplified and depicted heatmaps of identical colors for genes (Fig. 1b and 5a) and pathways (Fig. 1d and 5b) including the use of the same color scheme for human vs. mouse as suggested by the reviewer.”

Figures 2B, extended 2B, 6, 11 are still using different colours.

Response: We are now using the **same colors** for all heatmaps including Fig. 2B and extended Figs. 2B, 6, and 11. Note that, in Fig 2B and extended data fig 11, for clarity of the message only HCV vs Control is shown in a different color, while individually both the groups are represented in the same color.

- Human cohort data:

Page 5: “For the different patient cohorts (c, d, e, and f; Fig. 5a), however, a statistical analysis was not feasible as they followed different procedures for library preparation and RNA-seq.”

Page 7: “As pointed out in the response above RNA-sequencing of patient cohorts (cohort c-f) were performed with different techniques and platforms from collaborating centers. This precludes a ‘direct’ statistical comparison between them.”

If the data obtained with this different cohorts are not comparable, they should not be compared! Considering the previous comment, this part must be removed from the manuscript if now additional validation of the comparison is provided.

Response: We agree with the reviewer that the cohorts are not directly comparable as they address three different independent questions. To address this point and improve readability, we have modified and simplified the figure panel 5a, **by showing only the disease relevant rhythmic pathways with significant (FDR<0.05) and robust alterations in patients**. Moreover, we moved the results of the other three cohorts to the supplement as

independent figure panels since they cannot be compared directly with each other (**Extended Data Fig. 18a-c**). The corresponding text section has been modified accordingly.

Finally, we addressed the strengths and limitations of the study including its clinical translation in the discussion: *“A strength of our study is the use of two state-of-the-art model systems for HCV infection with data integration of liver tissues of HCV-infected patients. A limitation of the study is that these model systems only partially mimic virus-host interactions in patients and a diurnal analysis of the patient liver transcriptome and proteome is not available.”* (page 21, lines 11-15).

-SIRT1 and PPARG expression:

Page 16: “This approach validated our observation of an abrogated SIRT1 oscillation in human hepatocytes of HCV-infected chimeric livers.”

Authors provide new results showing that SIRT1 expression is lower at ZT12 compared to ZT0 in control animals while no time difference is observed in infected animals. With only 2 time points, it is misleading to discuss “rhythmicity”. In addition, no statistical analysis is provided for the comparison between control and infected animals. According to the observed variability, there is likely no difference and, if any, this could not explain the difference in acetylation. Altogether, this “SIRT1 story” is still not supported by the data.

Page 16: “Fig. 4g: Indeed, PPARG transcript levels are not significantly different between control and HCV-infected samples (page 17, lines 10-11). However, validation experiments using Western blotting and immunofluorescence confirmed that HCV-infection increases PPARG protein levels (4h-i and Extended Data Fig. 23). We describe these additional validations in the revised manuscript on page 17, lines 11-14.”

This is a cell experiment, completely unrelated to the question. PPARG must be measured in the animal models to justify this conclusion. This also must be the case for BMAL1 protein.

Response: We thank the reviewer for his/her critical analysis of our data. We agree that the PPARG results are completely unrelated to the question. To address this point, we have deleted the result of this experiment and are not following up on it.

We also agree with the reviewer that the SIRT1 *in vivo* data with only 2 time points and only a trend may be misleading to discuss “rhythmicity”. We therefore also deleted the results of this experiment.

We would like to explain why we used the HCV cell culture model: the HCV cell culture model (Wakita, Pietschmann et al. Nature Med 2005, Lindenbach et al. Science 2005) has

been shown to model key cell circuits driving liver disease and cancer in patients (Lupberger et al. Gastroenterology 2019, Hamdane et al. Gastroenterology 2019, Jühling et al. Gut 2021) and is the only available robust model to perform loss-of-function studies using RNAi or CRISPR-Cas9. A human-hepatocyte specific knock-down in human liver chimeric mice (HLCM) is technically extremely challenging since it requires adeno-associated virus gene transfer which has a poor transduction efficacy and **AAV-HCV cross-talk will not allow HCV-specific readouts**. Furthermore, since the HLCM liver is chimeric with mouse and human hepatocytes, species-specific protein analyses are technically challenging due to **cross-reactivity** of most of the commercially available antibodies across species.

Due to the limited remaining *in vivo* material (which was used for the extensive studies described in the manuscript) combined with the technical challenges of studying protein expression in human liver chimeric mice (most of the commercial antibodies against the human proteins have at least some cross-reactivity to the mouse orthologs resulting in absent or limited specificity for quantification), we were not able to perform additional protein analyses. A repetition of the HLCM study is not easily possible also due to ethical considerations (the experiment uses human tissues, has been performed already two times and the requested added additional information is limited for the overall conclusions of the study), its high costs and long duration.

We therefore addressed the reviewer's questions by down-toning or removing the conclusions of the "SIRT1 story". We now: (1) clearly indicate that SIRT1 and BMAL1 results were obtained in the cell culture model in both the revised text "*Given the technical challenges associated of performing specific loss-of-function studies in human hepatocytes of the chimeric mice, we utilized the state-of-the art cell culture system of HCV infection^{13,19,20,35,46,47} to investigate potential mechanistic links between SIRT1 and key CC gene BMAL1. It is of interest to note that this infectious system has been shown to model key cell circuits driving disease biology in HCV-induced fibrosis and cancer in patients^{19,20,35}.*" (page 16, lines 5-10) and "*These results may suggest a potential functional role for SIRT1 to regulate BMAL1 expression in the HCV cell culture model.*" (page 16, lines 17-19) as well as **legend of Fig. 4**; (2) removed the sentence describing SIRT1 from the abstract, (3) described the results of the cell-based experiments as a **potential candidate** mechanism of action among other mechanisms which will require further *in vivo* validation (e.g., on the human proteins and/or investigation such as *in vivo* loss-of-function studies) "*A strength of our study is the use of two state-of-the-art model systems for HCV infection with data integration of liver tissues of HCV-infected patients. A limitation of the study is that these model systems only partially mimic virus-host interactions in patients and a diurnal analysis of the patient liver transcriptome and proteome is not available.*" (page 21, lines 11-15) and "*In the state-of-the-art cell culture model, HCV reduced*

SIRT1 protein and *BMAL1* levels (Fig. 4) suggesting virus-induced perturbed *SIRT1* and *BMAL1* expression as a potential candidate mechanism contributing to the virus-induced changes of the rhythmicity of part of the transcriptome. Of note, the regulatory role of *SIRT1* on *BMAL1* has been previously established in mice^{45,57,58}.” (page 22, lines 2-6); (4) discussed the limitations of the *SIRT1* cell culture results and the question of *in vivo* relevance in the discussion. Our revised statement in the discussion reads:

“Interestingly, we also observed an HCV-induced perturbation of the promoter-enhancer associated histone mark (H3K27ac) *in vivo*, while gene-body associated H3K9ac was only disrupted during ZT8-ZT20 (Fig. 3a-h), indicating a dynamic interaction between different chromatin remodelers regulating gene expression at the epigenomic level. These alterations in different TFs, chromatin modifiers and their downstream targets suggest additional mechanisms mediating HCV-induced activation of pathways known to drive liver disease (Fig. 2-4 and Extended Data Fig. 20) to be explored further” (page 21, lines 16-22).

A potential candidate among others could be the NAD⁺-dependent deacetylase *SIRT1*⁵⁷. In the state-of-the-art cell culture model, HCV reduced *SIRT1* protein and *BMAL1* levels (Fig. 4) suggesting virus-induced perturbed *SIRT1* and *BMAL1* expression as a potential candidate mechanism contributing to the virus-induced changes of the rhythmicity of part of the transcriptome. Of note, the regulatory role of *SIRT1* on *BMAL1* has been previously established in mice^{45,57,58}. Supporting this hypothesis, we observed a perturbed expression of *BMAL1* targets in HCV-infected HLCM livers including TFs like *DBP* and several other genes (Extended Data Fig. 11b and Supplementary Table 4). However, additional *in vivo* validation and loss-of-function experiments would be required to unravel the detailed mechanistic events and cellular drivers in the diurnal regulation of disease pathways including the confirmation of a functional role of *SIRT1*.” (page 22, lines 1-11).

(Additional comments to the manuscript)

- Authors now admit that the *GH*, *STAT5*, and *mTOR* pathways lost rhythmicity, referring to Supplementary Table 3. There is absolutely no analysis of these pathways in this table. What leads to this conclusion?

Response: To address this point, we clarified in the revised manuscript that key genes of *GH*, *STAT5* and *mTOR* signaling were part of the dryR model 1 (non-rhythmic genes; Supplementary table 3) and now provide an analysis of genes in these pathways in the Supplementary Table 3. Our revised statement reads as follows: “Consistent with a recent study²⁷, we also observed that some of the key genes involved in growth hormone (*GH*)-*STAT5*³¹ and *mTOR* signaling pathways, e.g., *STAT1*, *STAT3*, *STAT5B*, *MTOR*, *LAMTOR1-5*

were non-rhythmic (*model 1*) in transplanted human hepatocytes (Supplementary Table 3)” (page 9, lines 13-16)

- *Figure 2d and associated text: This figure is supposed to show the “disruption” of rhythmic pathways, but most pathways are not rhythmic and show only in difference in the level of enrichment. This must be more clearly described.*

Response: We now clarified that all disease-relevant pathways depicted in Fig. 2d were previously determined to be cycling by dryR (Fig. 1d) within model 2 (Loss of rhythmicity). We clarified this in the text: “*Importantly, pathways (enriched for cycling genes; FDR <0.05) being significantly dysregulated comprise key drivers of chronic liver disease, including metabolic alterations (fatty acids, lipids, peroxisome organization), fibrosis (TGFβ-signaling, SMAD activity, fibroblast proliferation, and EMT response), and oncogenic pathways linked to HCC development (liver cancer signatures, MYC, H-RAS, and EGFR signaling) (Fig. 2d, Extended Data Fig. 13). All these pathways were previously determined to be cycling by dryR (Fig. 1d) within model 2 (Loss of rhythmicity) (Fig. 2d).*” (page 13, lines 6-13). During the previous revision, reviewer 6 requested to replace the JTK analyses in Fig. 1 by dryR. We made corrections in Fig. 2d and replaced some of the pathways, which initially being identified by JTK but not by dryR (GO Regulation of SREB by HM Fatty acid metabolism, GO Response to insulin by GO Peroxisome organization, GO TGFb2 production by GO Fibroblast proliferation, BC NFKb pathway by NFKAPPAB65_01, DASU IL6 scar up by HM Inflammatory response, BC RELA pathway by HOSHIDA Liver cancer survival down, CHUANG Oxidative stress response up by ACEVEDO Liver cancer up). We corrected the corresponding text section, accordingly (see this response above). Consistently, we re-calculated also the dysregulated pathways displayed in Fig. 5 and Extended Data Fig. 18 without causing major changes.

- *Statistics for the difference of rhythmicity are still missing for histone modifications.*

Response: We now provide a statistical analysis of the histone modifications in new **Figs. 3c and 3f** confirming the HCV-induced significant ($p < 0.05$, Mann-Whitney test) increase of H3K27ac and H3K9ac peak numbers at the transcriptional start site (TSS).

- *Information about the human hepatocyte donors is missing.*

Response: We now added additional information about the PHH donor in the revised manuscript: “*Humanized liver chimeric mice were produced by splenic injection of cryopreserved PHH (BD Biosciences, San Jose, CA) into uPA/SCID mice as previously described⁵⁷. The PHH donor (HF284) was a deceased 2-year-old female patient with no*

recorded history of liver disease and viral infection. The isolated PHH showed positive enzymatic activities for multiple human Cytochromes (CYP).” (page 32, lines 17-21)

- Repetitions of the descriptions of the models (pages 8 and 13).

Response: This has been addressed by **removing the redundant description of dryR models** (page 12, line 22, page 13, lines 1-3): “*Similar to Fig. 1c, we identified five distinct categories of transcriptional profile in HCV-infected HLCM livers (loss, gain, unaltered, and altered rhythm (Fig. 2c and Supplementary Table 4), plus one additional category of non-rhythmic genes in comparison to non-infected controls.*”

- Authors still refer to “circadian” rhythmicity throughout the manuscript.

Response: To address this point, we now systematically **removed** the term “circadian” in the context of rhythmicity from the manuscript. Nevertheless, given the use of term “circadian” in two recent studies published recently with human liver chimeric mice and human hepatocytes (Delbes et al., **Sci Adv** 9, eadf2982, 2023, and March et al., **Sci Adv** 10, eadm9281, 2024) and allowing to find our study by scientists and physicians interested in the circadian clock, we would prefer to keep the term *circadian* at least in the title (to be discussed with the editor in case of disagreement).

Reviewers' Comments:

Reviewer #6:

Remarks to the Author:

This new version of the manuscript by Mukherji and colleagues presents some significant improvements compared to the previous one. Nevertheless, it still contains several issues that need to be fixed before publication. Particularly, there is still several misleading statements despite authors claim “we now removed broad claims on perturbations of rhythmicity and adequately described results without overselling them”. Below is a detailed description of the remaining issues:

- Line 63-64: “perturbs the transcriptome by altering the rhythmicity of the expression of more than 1500 genes”. Number of genes in models 2, 3, and 5: $621+387+28= 1036$. This is not “more than 1500 genes”.
- Line 67: “these data support a role for virus-induced perturbation of the liver clock in cancer development”. As recognised by the authors, the circadian clock is globally not altered after HCV infection.
- Line 126: Authors wrote in the result section that data were analysed with Metacycle but the method section describes that DryR was used. Please correct.
- Line 149: “dryR revealed ~1,700 rhythmic protein coding orthologous genes in HLCM liver”: $824+1052+749+103= 2728$. Or authors mean “differentially rhythmic between mouse and human transcript”? This needs to be clarified.
- Line 173-175: “we also observed that some of the key genes involved in growth hormone (GH)-STAT5B1 and mTOR signaling pathways, e.g., STAT1, STAT3, STAT5B, MTOR, LAMTOR1-5 were non-rhythmic”. The fact that some genes in one pathway are not rhythmic doesn't seem sufficient to claim that the entire pathway is not rhythmic, but OK.
- Line 236-237: “Notably, analyzing two independent animal infection experiments showed overall perturbation in the expression of key CC genes”. One more time, 2 out of ~20 genes showing a change in phase is not sufficient to claim that the circadian clock is perturbed. The Figure 2b and extended figures 11a still show artificial fold changes that are not statistically significant. This is very misleading and should be removed.
- Line 238-239: “which affected the expression of its target gene transcription factor DBP”. This is false as Dbp is in model 4, meaning not altered rhythmic gene expression. Additionally, suggesting that change in expression level of Nfil3 at one time point (ZT4) could be due to REV-ERBa whose expression is not altered is also misleading. These are wrong statements not based on an honest analysis of the data.
- Line 243-246: “Taken together, these analyses indicate that by perturbing the expression of circadian TFs, HCV infection likely impacts the global gene expression of their downstream targets, e.g., BMAL1-activated (E-Box-containing), and NFIL3-repressed (D-Box-containing) genes.” In addition to the problems explained before, authors do not provide any evidence of a global alteration of the expression of BMAL1 and NFIL3 target genes. One more time, this is false and misleading.
- Line 246-247: “serum HCV RNA levels in HLCM displayed a modest diurnal rhythmicity with a peak in the active phase (ZT16-ZT20)”. This is not supported by any kind of statistical analysis.
- Line 301-305: “This revealed an overall suppression of H3K27ac levels in the regulatory regions of genes controlled by CC-output regulatory TFs (e.g., DBP, HLF, TEF, NFIL3; regulators of D-Box genes)

correlating with an overall dysregulation of their target pathways (bile acid metabolism, xenobiotic metabolism; Extended Data Fig. 13). While authors show that H3K27ac levels indeed decrease at the binding sites for these transcription factors, it is not clear if it is at genes involved in bile acid and xenobiotic metabolism. Moreover, extended figure 13 shows that xenobiotic metabolism and bile acid metabolism, as well as MYC targets all show a big majority of genes in model 4 (green on the figure). It is therefore difficult to understand how authors conclude that all these rhythmic pathways are disrupted after infection with HCV.

- Line 319-349: While this data in cell culture is of interest, it is difficult to link them to the rest of the manuscript. Sirt1 expression is not altered after HCV infection and Bmal1 rhythmic expression only shows a slight change in phase. The concluding statement “suggesting that decreased SIRT1 (deacetylase) levels not only leads to a reduction in BMAL1 levels but may also contribute (along with other chromatin regulators) to the HCV-induced epigenomic alterations” could be true in cell culture but does not reflect any of the data obtained in the mouse model.

- In opposition to the suggestion of the authors, the term “circadian” is properly used in the two cited articles showing changes in circadian rhythms in constant conditions (constant darkness or in vitro after cell synchronisation). It is not properly used in this article and should be changed.

Point-by-Point response to the reviewer:

Reviewer #6 (Remarks to the Author):

This new version of the manuscript by Mukherji and colleagues presents some significant improvements compared to the previous one. Nevertheless, it still contains several issues that need to be fixed before publication. Particularly, there is still several misleading statements despite authors claim “we now removed broad claims on perturbations of rhythmicity and adequately described results without overselling them”. Below is a detailed description of the remaining issues:

Response: We thank the reviewer for the critical review of our study and the helpful comments which allowed us to improve the manuscript. **We have now resolved all the remaining issues raised and modified the manuscript and figures accordingly.** To track changes, we provide a detailed point-by-point response below and marked changes in red in the revised manuscript as stated as well in the point-by-point response

Remaining Concerns:

1. *Line 63-64: “perturbs the transcriptome by altering the rhythmicity of the expression of more than 1500 genes”. Number of genes in models 2, 3, and 5: $621+387+28= 1036$. This is not “more than 1500 genes”.*

Response: As suggested, we have now corrected this statement in the abstract to “**perturbs the transcriptome by altering the rhythmicity of the expression of more than 1000 genes**” (page 4, lines 9-10).

2. *Line 67: “these data support a role for virus-induced perturbation of the liver clock in cancer development”. As recognised by the authors, the circadian clock is globally not altered after HCV infection.*

Response: As suggested, we have now corrected this statement in the abstract to “**these data support a role for virus-induced perturbation of the hepatic rhythmic transcriptome and pathways in cancer development**” (page 4, lines 12-14).

3. Line 126: Authors wrote in the result section that data were analysed with Metacycle but the method section describes that DryR was used. Please correct.

Response: As requested, we have now corrected the statement and removed the reference of Metacycle. The revised text reads “**The rhythmicity of CC-gene expression in the chimeric livers were analyzed by dryR**” (page 7, line10).

4. Line 149: “dryR revealed ~1,700 rhythmic protein coding orthologous genes in HLCM liver”: $824+1052+749+103= 2728$. Or authors mean “differentially rhythmic between mouse and human transcript”? This needs to be clarified.

Response: To clarify, these ~1700 rhythmic protein coding genes includes- model 2 (824 genes), model 4 (749 genes), and model 5 (103 genes). **Model 3 (1052 genes) was excluded from this group as they do not contain genes which are rhythmic in human hepatocytes** of the chimeric mice (**Fig. 1c**). The revised statement reads “**dryR revealed ~1,700 rhythmic protein coding orthologous genes (models 2, 4, and 5) in human hepatocytes of HLCM liver**” (page 8, lines 11-12).

5. Line 173-175: “we also observed that some of the key genes involved in growth hormone (GH)-STAT531 and mTOR signaling pathways, e.g., STAT1, STAT3, STAT5B, MTOR, LAMTOR1-5 were non-rhythmic”. The fact that some genes in one pathway are not rhythmic doesn't seem sufficient to claim that the entire pathway is not rhythmic, **but OK**.

Response: Thank you very much.

6. Line 236-237: “Notably, analyzing two independent animal infection experiments showed overall perturbation in the expression of key CC genes”. One more time, 2 out of ~20 genes showing a change in phase is not sufficient to claim that the circadian clock is perturbed. The Figure 2b and extended figures 11a still show artificial fold changes that are not statistically significant. This is very misleading and should be removed.

Response: We have now completely removed these figures and texts describing them.

7. Line 238-239: “which affected the expression of its target gene transcription factor DBP”. This is false as *Dbp* is in model 4, meaning not altered rhythmic gene expression. Additionally, suggesting that change in expression level of *Nfil3* at one time point (ZT4) could be due to *REV-ERBa* whose expression is not altered is also misleading. These are wrong statements not based on an honest analysis of the data.

Response: We have now completely removed these figures and texts describing them.

8 Line 243-246: “Taken together, these analyses indicate that by perturbing the expression of circadian TFs, HCV infection likely impacts the global gene expression of their downstream targets, e.g., *BMAL1*-activated (*E-Box*-containing), and *NFIL3*-repressed (*D-Box*-containing) genes.” In addition to the problems explained before, authors do not provide any evidence of a global alteration of the expression of *BMAL1* and *NFIL3* target genes. One more time, this is false and misleading.

Response: We have now completely removed these figures and texts describing them.

9. Line 246-247: “serum HCV RNA levels in HLCM displayed a modest diurnal rhythmicity with a peak in the active phase (ZT16-ZT20)”. This is not supported by any kind of statistical analysis.

Response: The original purpose of this experiment was to simply determine whether on the day of the sacrifice all the ‘infected’ mice had HCV. We have now corrected the statement “**We also measured viral load from the serum of HCV-infected animals (Extended Data Fig. 11)**” (page 12, lines 1-2).

10. Line 301-305: “This revealed an overall suppression of *H3K27ac* levels in the regulatory regions of genes controlled by *CC*-output regulatory TFs (e.g., *DBP*, *HLF*, *TEF*, *NFIL3*; regulators of *D-Box* genes) correlating with an overall dysregulation of their target pathways (bile acid metabolism, xenobiotic metabolism; Extended Data Fig. 13). While authors show that *H3K27ac* levels indeed decrease at the binding sites for this transcription factors, it is not clear if it is at genes involved in bile acid and xenobiotic metabolism. Moreover, extended figure 13 show that xenobiotic metabolism and bile acid metabolism, as well as *MYC* targets all show a big majority of genes in model 4 (green on the figure). It is therefore difficult to understand how authors conclude that all these rhythmic pathways are disrupted after infection with HCV.

Response: Agreeing with the reviewer, we have now modified the sentence. The revised text reads “**This revealed an overall suppression of H3K27ac levels in the regulatory regions of genes controlled by CC-output regulatory TFs (e.g., DBP, HLF, TEF, NFIL3; regulators of D-Box genes)**” (page 14, lines 15-17).

The previous statements on bile acid and xenobiotic metabolism as well as that of MYC targets have been removed.

11. Line 319-349: While this data in cell culture is of interest, it is difficult to link them to the rest of the manuscript. Sirt1 expression is not altered after HCV infection and Bmal1 rhythmic expression only shows a slight change in phase. The concluding statement “suggesting that decreased SIRT1 (deacetylase) levels not only leads to a reduction in BMAL1 levels but may also contribute (along with other chromatin regulators) to the HCV-induced epigenomic alterations” could be true in cell culture but does not reflect any of the data obtain in the mouse model.

Response: The revised manuscript now shows only in vivo data. **We have now completely removed these figures and texts describing them.**

12. In opposition to the suggestion of the authors, the term “circadian” is properly used in the two cited articles showing changes in circadian rhythms in constant conditions (constant darkness or in vitro after cell synchronisation). It is not properly used in this article and should be change.

Response: We have now **replaced** the usage of the term ‘circadian’ throughout the **main text** by ‘diurnal’, ‘rhythmic’ and/or ‘temporal’, in a context-dependent manner.